



# Path and site effects deduced from transfrontier internet macroseismic data of two recent M4 earthquakes in NW Europe

Koen Van Noten[1], Thomas Lecocq[1], Christophe Sira[2], Klaus-G. Hinzen[3] and Thierry Camelbeeck[1]

1 Royal Observatory of Belgium, Seismology-Gravimetry, Ringlaan 3, B-1180 Brussels, Belgium
2 French Central Seismological Office, University of Strasbourg, Rue René Descartes 5, 67084 Strasbourg Cedex, France
University of Cologne, Bensberg Erdbebenstation,Vinzenz-Pallotti-Stasse 26, D-51429 Bergisch Gladbach, Germany

*Correspondance to*: Koen Van Noten (koen.vannoten@gmail.com)

**Abstract.** The online collection of earthquake testimonies in Europe is strongly fragmented across numerous seismological agencies. This paper demonstrates how collecting and merging *"Did You Feel It?"* (DYFI) institutional macroseismic data strongly improves the quality of real-time intensity maps. Instead of using ZIP code Community Internet Intensity Maps we geocode individual response addresses for location improvement, assign intensities to grouped answers within 100 km² grid cells, and generate intensity attenuation relations from the grid cell intensities. Grid cell intensity maps are less subjective and illustrate a more homogeneous intensity distribution than the ZIP code intensity maps. Using grid cells for ground motion analysis offers an advanced method for exchanging transfrontier equal-area intensity data without sharing any personal's information. The applicability of the method is demonstrated on the DYFI responses of two well-felt earthquakes: the 8 September 2011 $M_L$ 4.3 ($M_W$ 3.7) Goch (Germany) and the 22 May 2015 $M_L$ 4.2 ($M_W$ 3.7) Ramsgate (UK) earthquakes. Both events resulted in non-circular distribution of intensities which is not explained by geometrical amplitude attenuation alone but illustrates an important low-pass filtering due the sedimentary cover above the Anglo-Brabant Massif and in the Lower Rhine Graben. Our study illustrates the effect of increasing bedrock depth on intensity attenuation and the importance of the WNW-ESE Caledonian structural axis of the Anglo-Brabant Massif on seismic wave propagation: seismic waves are less attenuated – high Q – along the strike of the massif but are stronger attenuated – low Q – perpendicular to this structure, especially when they cross rheologically different seismotectonic units that are separated by crustal-rooted faults.

**Keywords:** 'Did You Feel It', Data Exchange, Crowdsourcing, Grid cell, Shaking, Goch, Ramsgate, Sediment Thickness, Attenuation, Population





## 1 Introduction

The online collection of earthquake testimonies from 'Did You Feel It?' (DYFI) inquiries is an effective crowdsourcing tool for (real-time) intensity analysis of the area over which an earthquake was perceived (Atkinson and Wald, 2007; Wald et al., 1999; Wald et al., 2011). Macroseismic intensity assessment describes the ground

motion effect over a settled area and contrasts to instrumental intensity which measures the ground motion at specific sites where seismic instruments are installed (Sbarra et al., 2012a). In poorly-instrumented areas the people's earthquake perception provides important seismic information on the earthquake's ground motion. For moderate- to large magnitude seismic events, the high quantity of responses makes the DYFI intensity assessment a robust tool (Hough, 2013a; Wald et al., 2011), not only for tectonic earthquakes but also for non-earthquake

related felt or heard events such as e.g. (civil) bomb explosions, sonic booms, construction works, sport/concert events, etc. In a homogeneous, isotropic Earth, intensity decreases with increasing epicentral distance in a concentric pattern. However, various parameters such as earthquake magnitude, source mechanism, site effect and earthquake rupture directivity affect the intensity decay resulting in complex, non-concentric intensity patterns (e.g. Tosi et al., 2000). Macroseismology can reveal these intensity patterns and is therefore often used in

conjunction with studies on peak ground acceleration (Atkinson and Wald, 2007; Lesueur et al., 2012; Martin et al., 2015, Souriau, 2006; Worden et al., 2000), site effect analysis (Nguyen et al., 2004; Sbarra et al., 2012a), induced seismicity (Hough, 2014) and vulnerability assessment in civil engineering (García Moreno and Camelbeeck, 2013; Stefano and Chiara, 2014).

The DYFI website of the US Geological Survey (USGS) questions nationally and internationally felt

earthquakes. The advantage of the USGS DYFI system is that macroseismic data are collected systematically over the whole North American continent through a well-calibrated algorithm (Dengler and Dewey, 1998; Worden et al., 2000). In Europe, however, the situation is more complex: at least 32 seismological institutes in 22 countries manage and maintain their own DYFI inquiry (Fig. 1; see Appendix A for hyperlinks), sometimes more or less modified from the original questionnaire of Wald et al. (1999). Despite this high number of inquiries, only the

following institutes map DYFI results in real-time: the British Geological Survey (BGS), Istituto Nazionale di Geofisica e Vulcanologia (INGV), the French Central Seismological Office (BCSF), the European-Mediterranean Seismological Centre (EMSC) and the Royal Observatory of Belgium – Erdbebenstation Bensberg network (ROB-BNS). The Royal Observatory of Belgium (ROB) shares its DYFI inquiry with the Erdbebenstation Bensberg (BNS) of the University of Cologne (Germany) to collect transfrontier macroseismic information across

the Belgian-German border (ROB-BNS network; Lecocq et al. 2009). In the ROB-BNS questionnaire, respondents are automatically forwarded to the BCSF if an earthquake is located in France and consequently macroseismic data are shared in real-time between these institutes. Germany has no organised national inquiry which results in eight different institutes gathering macroseismic data. Other national inquiries are often known to local population but unfortunately their macroseismic results are not shown in real-time.

The benefit of having different European inquiries is that every institute has made the inquiry available in its own national language(s) and hence can deal with specifics in their language in the open comment box. However, important disadvantages are that (1) people can answer to multiple national or international inquiries when they felt an earthquake and that (2) the data of the perception of transfrontier-felt seismic events is strongly fragmented across different institutes and countries. Merging databases carries the risk of duplicate entries in the

database but, more importantly, also has the risk of merging intensities that may slightly differ from questionnaire



to questionnaire or country to country due to a different intensity calculation procedure which will smooth the mean of the merged intensities. Because of this European fragmentation, performing a proper macroseismic assessment of transfrontier-felt earthquakes remains a complex and not straight forward task. Contemporary transfrontier macroseismic maps are either based on historical reports (e.g. Camelbeeck et al., 2014; Cara et al.,

2005; Knuts et al., 2015; Kronrod et al., 2013) or on communal reports, such as used to map the 1992 $M_L$ 5.8 ($M_W$ 5.3) Roermond earthquake (NL) and the 2003 $M_L$ 5.4 ($M_W$ 4.9) Rambervillers (FR) earthquake (Cara et al., 2005). These successful efforts in (historical) macroseismology indicate that a comprehensive method is needed to merge the scattered online DYFI data; however, in NW Europe, such a methodology is currently not implemented.

**FIGURE 1**

In a DYFI inquiry, it is essential and obligatory for the respondent to provide a ZIP code from the community where the earthquake was felt (Wald et al., 1999). This allows constructing ZIP code Community Internet Intensity Maps (ZIP-CIIM's) from which the geometric spreading of the macroseismic intensity can be evaluated. For earthquakes with a sufficiently large magnitude to be felt in numerous communities, the ZIP-CIIM provides usually a good overview of the affected area (e.g. Horton et al., 2015; Hough, 2012) and give valuable

information on the intensity decay with increasing epicentral distance. In many European countries ZIP code areas are small in densely populated areas and large in remote areas. A ZIP-CIIM has thus the advantage that the number of potential respondents per ZIP code can roughly be the same. The big downside of using ZIP-CIIM's, however, is the irregularity of the municipality shapes resulting in an inhomogeneous spatial coverage of intensities over the felt area.

Another common flaw in DYFI analysis is that epicentral intensities are often underestimated (Sira et al. 2016). Intensities at large epicentral distance, on the other hand, are often overestimated because people who have not felt the earthquake usually do not rush to the Internet to find information and share their experience. This usually results in a lack of intensity I (non-felt) reports. These not-felt responses are however important as they prevent the average intensity in a community being too high at large epicentral distance. The underrepresentation

of low intensities is not just an effect of population size within a ZIP-code because the number of responses in a community usually scales with the population size (Boatwright and Phillips, 2013; Mak and Schorlemmer, 2016). Currently, INGV, ETH and EMSC deals with this problem by reaching out to volunteers after an earthquake occurred near their location to request their (non-)felt experience. This allows to distinguish between intensities I, II and III and to avoid intensity overestimation (Sbarra et al., 2010, Bossu et al., 2016).

In this study, intensities are processed by grid cell mapping which ensures a more homogeneous spread of intensities than in a ZIP-CIIM. In our procedure, the address of individual intensity data points (IDP's) are first geocoded into their geographical coordinates and, second, an intensity is assigned to equally-sized grid cells covering the affected area. Because cell size is large enough (100 km$^2$) to capture enough responses, this method allows merging IDP's collected by different institutes of which the respondents' location may be imprecise. The

applicability of the method is demonstrated on two particularly well-felt, but without any reported damage, moderate-magnitude earthquakes that struck the NW European territory: the 8 September 2011 $M_L$ 4.3 ($M_s$ 4.1, $M_W$ 3.7) Goch earthquake (DE) and the 22 May 2015 $M_L$ 4.2 ($M_W$ 3.7) Ramsgate earthquake (UK). Both earthquakes caused non-circular distributions of intensities across NW Europe and their perception in Belgium, Germany, The Netherlands, Luxemburg, France and the UK definitely classifies them as 'transfrontier'. For each

earthquake, all institutional DYFI responses were collected and merged. This study provides a post-seismic (not





yet applied in real-time) methodology how European macroseismic datasets can be successfully merged and demonstrates why the studied events had non-circular intensity distributions taking into account source mechanism, population density, local geology, bedrock depth and intensity distributions of past earthquakes.

## 2 Methodology

### 2.1 Geocoding

Because intensity is determined for each ZIP code, ZIP-CIIM's cover irregular shapes on maps causing the intensity locations to be imprecise. In a macroseismic questionnaire, people are asked to provide their ZIP code to generate to ZIP-CIIM but they are free to provide their street address. (i) To improve the spatial resolution of macroseismic data and (ii) to provide a more realistic spread than in ZIP-CIIM's, the respondent's location (address) is first geocoded to its true geographical coordinates. The USGS has the capability to manually trigger a geocoding in-house algorithm to convert individual addresses into longitude and latitude coordinates by using publically available GPS geocoding services (Wald et al., 2011). However, their in-house algorithm to convert addresses automatically is not publically available. Exploring the address list in the macroseismic ROB-BNS database shows that people frequently abbreviate their street address to their convenience (e.g. av., av, ave = avenue; str, st = street; °, nr, N° = street number) and regularly make accidental typographical misspells. Although many official geocode systems such as postal services, official cadastre plans, and governmental services provide very accurate locations, these geocoding services usually require an exact and correct address input of the respondent and often cannot deal with abbreviations and typographical mistakes.

**FIGURE 2**

To account for this inconvenience, a straightforward algorithm (Fig. 2) was developed to send addresses automatically to a Google™ JSON Api to geocode the data. The free edition of the Google JSON Api is currently limited to 2500 address conversions per day which currently limits using Google for real-time geocoding of large number of responses to large-magnitude earthquakes (unless a business account is bought). We prefer the use of Google above the official services because it corrects addresses with typographical errors and the precision of the geocoded location is weighted by the following four quality factors:

1. *rooftop* quality = precise location to the street address;
2. *range interpolated* and *geometric centre* quality = location is as good as the street location
3. *approximate* quality = address could not be located. Either the ZIP code centre or a community location near the address is returned after geocoding.

As the *approximate* quality is too uncertain to be used, any address that has this quality was rejected from the geocoded databases in this study (Fig. 2). Depending on the user's needs, different quality factors and thus different subsets of the database can be selected for macroseismic intensity assessment. For a detailed intensity evaluation of a local site, for instance, only the *roof top* quality could be used to ensure a correct location. After geocoding, we automatically calculate the epicentral distance from the precise location of the observer to the epicentre obtained by instrumental data, which facilitates the evaluation of intensity distance attenuation. Analysing the address columns in the ROB-BNS (31,929 individual testimonies since 2002) and BCSF (~80,000 individual testimonies since 2000) macroseismic databases shows that ~89% and ~76% of people, respectively, freely give their personal address. The ROB-BNS geocoding system was first applied on 301 inquiries (235





geocoded) submitted by people that reacted to an air blast caused by the accidental explosion of a WOII bomb at 3 January 2014 at Euskirchen (Germany) (Hinzen, 2014). Although a geocoded map uses fewer responses and presents a subset of the data, the distribution of IDP's significantly improved the interpretation of the macroseismic field compared to ZIP code intensities.

**2.2 Intensity rescaling**

In the USGS DYFI system, based on the given answers in the questionnaire, , and Wald et al. (1999). The questionnaire used by the ROB-BNS network (see Lecocq et al., 2009) is largely based on the standardised questionnaire of Wald et al. (1999) using standard statistical procedures and rejections following Dengler and Dewey (2000). The questions concern typical effects ranging from transient effects to building damage and a decimal community intensity (CDI) value is determined by a computer algorithm. In the ROB-BNS database, first, the obtained intensities were manually checked to delete duplications and to avoid unrealistic answers that lead to unrealistic high intensities for the given earthquake magnitudes. Afterwards, a floor-level correction was applied for people living at higher floors in a multi-story building to avoid intensity overestimation due to wobbling of upper floors (prescribed in the EMS-98 guideline, p. 29, Grünthal et al., 1998). Following the EMS-98 guideline and Sbarra et al. (2012b), all intensities from responses at level three and four were decreased by one intensity value (i.e. -I) with a minimum value of intensity II to indicate that the earthquake was still felt. Responses in basements and above floor-level four were not taken into account. Entries from people living at floor levels zero, one and two were not rescaled.

**2.3 Intensity grid cell analysis**

In a geocoded dataset, IDP's that are statistically too high or too low (because the respondent did not answer all questions correctly) are not ruled out and may lead to slightly over- or underestimated intensities. Because of these artefacts, the distance decay of a geocoded dataset has a larger scatter than the distance decay of averaged intensities from a ZIP code dataset (Haase and Park, 2006). To account for this statistical problem, Wald et al. (1999) subdivided ZIP codes by regularly-sized grids of a few kilometres, i.e. smaller than an average community polygon, for which more realistic average intensities can be calculated. In this work, a similar grid cell approach is followed: after geocoding all IDP's, we first structure the model area where the earthquake was perceived in $100 \text{ km}^2$ (10 km by 10 km) grid cells. This cell size can be considered as equivalent to a Macroseismic Data Point (MDP), which is used in historical seismology to group IDP's and which in average has the size of a small city, according to the AHEAD guidelines. Second, the mean intensity of all (geocoded if possible) answers within a cell is determined. To avoid statistical errors, an intensity value is only assigned to those cells that have a least three responses (Fig. 2). The grid cell calculation is performed in QGIS (QGIS, 2016).

**3 Seismotectonic framework**

**3.1 The 2011 Goch earthquake in the Lower Rhine Graben (LRG)**

The Goch earthquake occurred on Thursday 8 September 2011 at 19:02:50 UTC (21:02:50 local time) close to the German-Dutch border near the city of Goch (population of 34k, DE). The nearest bigger city is Nijmegen (pop. 160k, NL), 25 km NW of the epicentre. Waveform analysis of seismograms recorded by different seismic



stations within the ROB-BNS network resulted in an epicentre location of 51.668°N ± 2.7 km, 6.162°E ± 2.4 km (Fig. 3) and a depth estimation of 10 km ± 5.9 km. The depth uncertainty is due to poor epicentral station coverage. Instrumental magnitude was determined at $M_L$ 4.3 ± 0.3 and $M_W$ 3.7 ± 0.3. The Goch earthquake occurred at the NE lateral end of the LRG, i.e. the most seismic active tectonic graben structure in NW Europe. The LRG is

defined by NW-SE oriented normal faults that are the cause of moderate seismicity (Vanneste et al., 2013). The dominant seismic activity within the LRG is clustered in the southern and southwestern part (Fig. 3). The LRG forms an asymmetric horst and graben structure of which the main graben is the Roer Valley Graben. Its eastern boundary is defined by the Peelrand boundary fault, along which the $M_L$ 5.8 ($M_W$ 5.4) 1992 Roermond earthquake occurred (Camelbeeck and van Eck, 1994). The Venlo Graben (VG in Fig. 3) is a smaller half-graben situated NE

of the Roer Valley Graben and is bordered in the NE by the Viersen Fault (VF in Fig. 3): a Quaternary active, ~53 km long continuous fault mapped from the German-Dutch border in the NW to Monchengladbach in the SE.

The focal mechanism of the Goch earthquake from the P-wave polarities of stations within the ROB-BNS network shows a left-lateral strike-slip faulting mechanism (Fig. 3; Strike: 354.3, dip: 76, rake: 1.7). Based on a mean fault orientation of N°332 (Vanneste et al., 2013) for the Viersen fault, the N°354 nodal plane

corresponds with the true fault plane and the E-W nodal plane as the auxiliary plane. The link between seismicity and the faults mapped at the surface or inferred faults at depth in the LRG is in many cases not obvious due to imprecise depth location or due to the stepping character of some faults (Hinzen and Reamer, 2007). Taken into account the NW dip direction of the Viersen fault, the 6 km ± 2.7 km distance of the Goch epicentre to the surface trajectory of the Viersen fault, and a 10 km source depth, it is thus not possible to attribute the Goch earthquake

to the Viersen fault. A more likely source is a fault strand parallel to Viersen fault, as mapped in Ahorner (1962). The Goch earthquake is remarkable because of its isolated position in the NE part of the LRG and Venlo Graben, and because it occurred without any measured aftershocks. Catalogues list only two small earthquakes within a radius of 20 km around the Goch epicentre: the $M_L$ 2.8 event near Brandemolen (DE) on 4 April 1997 and the $M_L$ 0.9 near Venlo (NL) on 21 May 2007. The left-lateral strike-slip mechanism is consistent with a maximum

horizontal compressional stress oriented N°310, an orientation within the range of the NW-SE oriented regional stress tensor variation within the LRG (Camelbeeck and van Eck, 1994; Hinzen, 2003).

**FIGURE 3**

### 3.2 The 2015 Ramsgate earthquake in the Anglo-Brabant Massif (A-BM)

The Ramsgate earthquake occurred on Friday 22 May 2015 at 01:52:17 UTC (02:52:17 local time) with an

epicenter three km offshore the eastern coast of the United Kingdom in the Strait of Dover. The closest town is Ramsgate, a small coastal town (pop. 40k) in the County of Kent, SE of London. The Ramsgate earthquake was located at 51.304°N, 1.438°E according to the BGS (Fig. 3) and estimated at a depth of 15 km. This location corresponds well to the 51.283°N, 1.409°E location derived from the ROB-BNS network. Instrumental magnitude was determined at $M_L$ 4.2 according to the BGS. Focal mechanism is an oblique thrusting mechanism along a NE-

SW thrust fault (Fig. 3) with a small extent of a ruptured area (~1 km$^2$).

Small to moderate-sized earthquakes are not unusual for the low- to moderately-active seismic zone of the Strait of Dover/Pas de Calais (Musson, 2004). The latest felt event in this region was the 28 April 2007 $M_L$ 4.3 ($M_W$ 3.9) Folkestone earthquake located 27 km SW of Ramsgate causing small damage in the Foord district at Folkestone (Ottemöller et al., 2009; Ottemöller and Sargeant, 2010). Other larger magnitude events that caused

strong ground motions and damage in the UK, France and Belgium are the historical earthquakes of 21 May 1382





($M_S$ 6.0) and 23 April 1449 ($M_S$ 5.5) and 6 April 1580 (M ~ 6), rather poorly located by macroseismic analysis (Melville et al., 1996) but felt up to ~400 km epicentral distance on the continent (Camelbeeck et al., 2007). Given their depths, the sources of the Folkestone, Ramsgate and the moderate-magnitude North Sea earthquakes are expected along Caledonian faults in the basement rocks of the A-BM: the Lower Palaeozoic massif north of the

Variscan Front that extends from a WNW-ESE orientation in the southeastern part of the UK to central and North Belgium to ENE-WSW orientation in eastern Belgium (Fig. 3). Seismicity within the seismotectonic zone of the A-BM is considerate low to moderate (Camelbeeck et al., 2007), however, the largest onshore earthquake ever recorded on the Belgian territory, i.e. the Zulzeke-Nukerke 11 June 1938 $M_s$ 5.0 earthquake (Fig. 3), occurred within the A-BM, indicating that large magnitude earthquakes can occur.

**4 Results: Transfrontier macroseismic intensity assessment**

**4.1 Integrating macroseismic databases**

The 2011 Goch event was widely felt in Germany, Belgium, Luxemburg and The Netherlands. Macroseismic data was collected by the ROB-BNS network, by the Royal Netherlands Meteorological Institute (KNMI) in The Netherlands, and by the Geological Survey of Nord-Rhein Westfalen (NRW-GD) in Germany, a network that

overlaps with the ROB-BNS network. Felt reports for the 2015 Ramsgate event were collected at following seismological institutes: the ROB-BNS network for the Belgian territory, the BCSF for the French territory and the BGS in the UK. Apart from these national inquiries, also the international EMSC and USGS agencies assembled DYFI witness reports of the two earthquakes (Tables 1 and 2).

         To fully understand the effect of seismic radiation across the national borders of countries where the

Goch and Ramsgate earthquakes were perceived, it was crucial to collect and merge all available macroseismic information from these international agencies. In the months after the events, the macroseismic data were freely available upon request from all institutes, after the personally identifying information was stripped. To protect the location of the respondent, the USGS provided a list in which the geographical location coordinates were truncated up to two decimal digits. The precision of the response is therefore limited within a range of 1.11 km, causing

some responses to overlap on the map. Also the NRW-GD protected the location data up to three decimal digits with a location error of 0.11 km. Although this truncation limits the response precision, it does not pose a problem for intensity analysis as all available macroseismic data is grouped within 100 km² cells, which are much larger than the individual location error.

**4.2 Intensity assessment**

In the US, intensity assessment within a CIIM is calibrated to be, on average, similar to the Modified Mercali Intensity (MMI) (Musson et al., 2010). Within Europe the MMI is translated to the European Macroseismic Scale 98 (EMS-98) (Grünthal et al., 1998). Intensities assigned by these and other modern scales are found to correspond closely to each other (Hough, 2013a). The results of the merged online DYFI inquiries of the different institutes are mostly based on the questionnaire of Wald et al. (1999), however, with modifications. In the ROB-BNS,

KNMI, BGS, EMSC and BCSF agencies, analysis of the questionnaire is based on the EMS-98. The impact of differences between the institutional questions on the intensity scale remains very low (except for the NRW-GD questionnaire, see further). The differences between institutes are more visible in the number and spatial variability





of the collected data. Hence, merging the networks of the subscribed informants allows having more data, removes the spatial variability and ensures a better reliability of the macroseismic information. Examples of other transfrontier macroseismic maps based on online testimonies, e.g. 2003 Rambervillers (FR), 2004 Waldkirch (DE), all show a good equivalence in intensities at the borders which justifies merging databases.

5       After database request, ROB-BNS, KNMI, BCSF and USGS intensities were provided in CDI with one digit. The NRW-GD uses a different approach for intensity calculation and only uses half intensity units. EMSC and BGS macroseismic data were provided in integers. In what follows, grid cell intensity values are either noted in decimals (e.g. a CDI of 3.7) or in Roman numbers in which intensities are rounded between x.5 and 1+x.5 (e.g. a CDI of 3.7 results in intensity IV). Cells with a mean CDI of $1 < CDI < 2.5$ are coloured into intensity II

10  indicating that the earthquake was felt in that cell (Fig. 2).

### 4.3. The $M_L$ 4.3 Goch earthquake

**Table 1**

**FIGURE 4**

**FIGURE 5**

### 4.3.1 ROB-BNS

The ROB-BNS database (3294 responses) shows the largest geographical spread, mainly because the questionnaire is provided in four languages (see Appendices A & B) allowing to survey far across the national borders. A total of 2307 addresses (69%) of the ROB-BNS database could be geocoded. This rather low percentage is mainly because a large amount of German responses lacked street address information. After floor-level correction 2134 responses (65%) remained for intensity assessment (Table 1). Floor-level correction affects the grid cell intensities of large cities: Brussels and Verviers in Belgium and Cologne, Düsseldorf, Wuppertal and Essen in Germany as they all rescaled from intensity III to intensity II. Intensity IV results for all cells in an area of 30 km around the epicentre. Maximum CDI's are 4.4 and 4.1 in the cells 15 km south and 15 km north of the epicentre, respectively. Epicentral CDI is 3.7. The ROB-BNS macroseismic distribution is underrepresented NW of the epicentre in The Netherlands at larger epicentral distance (Fig. 4).

### 4.3.2 KNMI

The KNMI questionnaire contains 4222 responses; 3638 (86%) locations were geocoded to street level or better. As the KNMI enquiry is only provided in Dutch language, almost all responses (except from five entries at Emmerich am Rhein in Germany) were submitted from in The Netherlands (Fig. 4). After floor-level correction 3637 responses remained for intensity analysis. The intensity of only 14 responses needed to be rescaled. Intensity rescaling did not lead to intensity decrease of any grid cell, even not in major cities such as Nijmegen, Utrecht, Rotterdam or Amsterdam. Intensity IV is assigned to three cells to west and NW of the epicentre located south of Nijmegen (NL). An epicentral intensity cannot reliably be calculated due to a lack of data points (only three) in the epicentral grid cell.





### 4.3.3 NRW-GD

To 1149 entries of the NRW-GD questionnaire, a proper intensity could be assigned. As the questionnaire is only provided in German language, almost all responses, expect one from Heijen (NL), were submitted from in Germany. NRW-GD did not share addresses but only provided truncated, three digits individual coordinates. Due to the large 100 km$^2$ cell size, locations are still properly grouped within the correct cell. Intensity data could not be floor-level corrected as this question is missing in the questionnaire. Remarkably, the NRW-GD database lacks a single not-felt report, intensity I is completely lacking in their distribution (Fig. 5). The half intensity units, the missing floor-level question and the lack of intensity I reports strongly affects the intensity distribution in the grid cells (Fig. 5) with respect to the other institutes. In the German affected area, NRW-GD grid cell intensities are systematically higher (on average a half unit but sometimes more than one unit) than the grid cell intensity of ROB-BNS. Grid cell intensity in Düsseldorf, for example, was rescaled after floor-level correction to CDI 2.5 in the ROB-BNS database whereas in the NRW-GD the intensity is CDI 3.3. There are also many isolated grid cells, sometimes 100 km from the epicentre (e.g. Wuppertal, Dortmund) that still show intensity IV although in the ROB-BNS database intensity III is assigned to these cells, or even are floor-level corrected to intensity II.

### 4.3.4 EMSC

At the EMSC, macroseismic data are collected in two different ways: (1) through a classic questionnaire, (2) via a list of thumbnails (cartoons) describing shaking level (Bossu et al., 2008). The information gathered from the questionnaires is divided in an optimal number of clusters and a representative intensity is assigned to each cluster (Amorèse et al., 2015). At least five responses are needed in order to assign an intensity to a cluster. The cluster data is comparable to a grid cell, although clusters may have different sizes. The floor-level question is on the EMSC web form but is not taking into account in the intensity estimation.

A total of 259 people reported to the EMSC via the thumbnails and 269 via the questionnaires. Plotting individual thumbnail locations on a satellite map shows that the locations are often not geocoded for *rooftop* quality. Most responses came from The Netherlands and Germany. Because of the low amount of data, dividing the affected area in grid cells results in a fragmented image of the affected area (Fig. 4). In the epicentral area, intensity IV is never reached. This contrasts with the clustered data of the EMSC in which intensity V was assigned to the clusters of the epicentral area. Given this discrepancy, we only used the thumbnail information in the final intensity assessment as these correspond better with the other institutional data.

### 4.3.5 USGS

After collecting individual responses, the USGS assigns a weighted value to the answers of different questions (i.e. the felt index, shelf index, damage index, etc.). The responses are then averaged to reflect a consensus of the grouped answers. We used the weighted, one digit, CDI intensities of 1038 people that reported to the USGS to have felt the Goch earthquake. Personal details were protected by truncating location coordinates to two digits. The USGS grid cell intensities are similar to the ROB-BNS intensities, despite the low amount of grid cells on the USGS grid cell map (Fig. 4). A floor-level correction could not be applied due the missing question. Yet, most people reported from The Netherlands to the USGS. Rescaling the KNMI data for floor level hardly affected grid cell intensity due the low amount of reports from multi-store buildings. Therefore using USGS intensity data is



appropriate, despite absence of floor-level correction. USGS grid cells show an epicentral intensity of IV. Maximal CDI is 4.1 in the cell north of the epicentre.

### 4.3.6 Comparing and merging databases

With respect to the epicentre, the 2011 Goch earthquake was felt from Amsterdam (NL) in the NW, Paderborn (DE) in the East, Darmstadt (DE) in the SE to Brussels (BE) in the SW. For these four countries involved, **10,281** people responded to the online questionnaires of six institutes (Table 1). The furthest submitted response came from Lille (FR) at 245 km. After geocoding or by using the truncated location provided by the institutes, 8218 entries could be used for intensity analysis. Before merging macroseismic data of all institutes, first the quality of the different macroseismic sources are evaluated. The distribution histograms (Fig. 5) show the individual and grid cell intensity distributions of all entries and for each institute separately, binned by half intensity units. The number of cells in each histogram corresponds to the same amount of cells in Figures 4 and 6. Cells with less than three responses are not included in the grid cell distribution. Several observations can be made when comparing the distribution of the individual entries with the grid cell histogram:

- Although high intensity individual IDP's exist, merging IDP's in grid cells and averaging their intensity reduces the influence of overestimated intensities;
- Grid cell intensity assessment results in a normal distribution of the macroseismic population whereas such a distribution is lacking in the individual database, no matter which institutional data is considered. This is a plausible result given the fact that the area to which intensities II and III are assigned is larger than the epicentral area to which intensity IV is assigned. It also shows that intensity I has been systematically underreported.
- Merging intensities from national agencies results in a denser coverage of the felt area (Fig. 4) and higher number of responses (Fig. 5) than the international USGS or EMSC inquiries alone.
- The mean intensity of the grid cell distribution of transfrontier inquiries (ROB-BNS, EMSC and USGS) is lower (I = 2.5) than the mean intensity of national inquiries (I = 3 for KNMI and I = 3.5 for NRW-GD). It seems plausible that the grid cell distribution of NRW-GD database is higher than KNMI because more responses are gathered from the epicentral area in Germany, whereas KNMI has no entries from this area.
- The NRW-GD received many responses, some of which were located more than 150 km from the epicentre. Yet, intensity I (non-felt) responses lack in their database. Because intensity data could not be floor-level corrected, intensities of cities are mostly higher than in the database of other institutes. Because of this, the NRW-GD intensity distribution is shifted towards higher intensities compared to other institutional distributions (Fig. 4). Based on these comparisons it is concluded that the NRW-DG intensities are systematically overestimated. To avoid intensity overestimation, NRW-GD data were further not used when databases are merged for intensity mapping and attenuation but are used to discuss the felt impact of the Goch earthquake.

After geocoding, floor-level correction and exclusion of NRW-GD data, 7068 entries remained for final intensity assessment. 663 people reported to have not felt the earthquake (intensity I). An intensity could be assigned to 309 grid cells (Fig. 6B), 24 cells were not used as they contain only NRW-GD data. The timing of submissions is spread between a few minutes to three days after the event. Evaluating the location and timing of submission





shows that entries were random in time and space and that regional and national news flashes did not cause any centralisation of responses.

**FIGURE 6**

The Goch intensity pattern is far from a circular radiation from the source (Fig. 6). Intensity IV is
distributed in a NW-SE direction, parallel to the main trend of the LRG and the Venlo graben. Cells with intensity
IV are all limited to the NE of the epicentre, an area that corresponds to the horst structure of the Venlo Graben.
The felt effect is clearly distributed in a NW, N, NE and SW pattern, but observations are lacking in the SW part
of The Netherlands and in NE Belgium. However, the Goch earthquake was felt in central Belgium from Liège
up to Brussels. Also in the SW extend of the LRG in Germany, i.e. the area W of Cologne and E of Aachen, the
Goch event was mostly not perceived, resulting in an absence of grid cell intensities in that area.

### 4.4.7 Attenuation analysis

In our analysis, all macroseismic information has been grouped in the grid cells. Hence, performing an attenuation
analysis by using grid cell information leads to an equal-area intensity attenuation relation (IAR) that is not biased
by concentration of responses due to population density. In Figure 7 the epicentral distance of each IDP and each
centroid, i.e. the geographical centre of a grid cell, is plotted versus the intensity of that observation/cell. Similar
to the grid cell maps, only centroids with at least three responses are used in the attenuation graph. To evaluate
the mean distance decay, we use a moving bin technique in which the mean intensity is calculated for a bin width
of 20 km which moves in overlapping steps of 2 km. Using the mean value of moving bins has the advantage of
deriving a smoother attenuation curve than using a stepping bin technique, in which the bin moves with steps
similar as the bin width.

**FIGURE 7**

The Goch IAR is both derived from individual IDP's (Fig. 7A) and from grid cell intensity data (Fig.
7B). Although population density strongly biases the distribution of individual responses (significant more
responses in the first 40 km, see horizontal histogram in Fig. 7A), the IAR is not affected by it as the shape of
both mean curves is similar in the first 50 km. Several studies on historical macroseismology have provided
regional attenuation intensity relations applicable for NW Europe (e.g. Backun and Scotti, 2006; Hinzen and
Oemisch, 2001; Stromeyer and Grünthal, 2009). IAR's derived from historical data, however, always suggest a
more widespread felt effect than the spatially-rich DYFI data (Hough, 2013a, b). We therefore compare our
derived IAR's with the CEUS attenuation prediction model of Atkinson and Wald (2011), applicable for intraplate
earthquakes, because it is the only one that has been derived from online DYFI data alone. In Figure 7, CEUS
predicted intensities are illustrated for an $M_W$ 3.7 earthquake, i.e. the equivalent of an $M_L$ 4.3 event following the
Reamer and Hinzen (2004) conversion for the LRG.

The epicentral intensity ($I_0$) of the 2011 Goch earthquake is unusually low. The epicentral centroid has a
CDI of $I_0 = 3.7$, which is one intensity unit lower than the CEUS predicted intensity. In the first 50 km, intensity
decays with one intensity unit, probably related to the fast energy decay by increasing distance from the source.
Between 50 km and 90 km, the IAR flattens around intensity III. This amplifying effect is typically related to
post-critical reflections from the Moho discontinuity that join direct seismic arrivals (Atkinson and Wald, 2007).

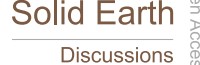

The slightly larger distance of the Moho bounce arrival in the CEUS prediction model (80 km) than in the Goch model (65 km) is probably related to the more shallow Moho in NW Europa, especially below the LRG.

A continuous intensity decay is observed between 100 km and 140 km, however, at a slower rate than in the first 50 km. This change in slope is related to the underreporting and lack of intensity I reports in the database in this distance range. The underreporting is visible in the histograms (Fig. 7B) where the frequency of centroids strongly decreases after 100 km, although an increase should be expected because more people (and more grid cells) were affected by the earthquake at this distance range.

**FIGURE 8**

At a distance of 165 km, 111 responses were submitted by people that felt the earthquake at Brussels (pop. 137k). Despite this low percentage (less than five responses per 10,000 inhabitants; Fig. 8B), this response is still considerably larger than those from many other populated cities at closer epicentral distance at which the Goch event was not perceived. In Belgium, no responses were submitted from the cities of Hasselt (91 km, pop. 56k) or Mechelen (135 km, pop. 117k), and only two and three were submitted from Turnhout (93 km, pop. 50k) and Antwerp (125 km, pop. 600k), respectively. In fact, whole Flanders (northern Belgium) did hardly report the earthquake, although it concerns a densely populated area (Fig. 8A).

Between 140 km and 175 km epicentral distance, i.e. the furthest grid cells to which a reliable intensity could be assigned, a small intensity increase is given by the IAR of the grid cell data (Fig. 7B). This increase exemplifies an intensity amplification that is caused by the grid cells located between Liège (azimuth of 197°) and Brussels (azimuth of 237°). Remarkably, people from the cities of Assen (150 km, pop. 69k) in The Netherlands and Osnabrück (145 km, pop. 196k), Bielefeld (168 km, pop. 327k), Siegen (153 km, pop. 108k) and Koblenz (175 km, pop. 110k) in Germany did not report to have felt the earthquake, although they are at closer or similar distance than those living at Brussels. The lack of responses from these large cities shows that the amplifying shaking effect in the IAR between 140 km and 170 km is caused by the Belgian macroseismic data and is clearly not a matter of population density.

### 4.4 The 22 May 2015 Ramsgate earthquake (UK)

Five institutes received macroseismic data for the Ramsgate earthquake: i.e. BGS, EMSC, USGS, BCSF and ROB-BNS. Although the Ramsgate earthquake occurred at 02:52:17 local UK time in the morning, it was widely reported in the UK, France, Belgium and, to a low extend, in The Netherlands on the continent. The Ramsgate event ($M_L$ 4.2) was perceived unusually far on the European continent, up to an epicentral distance of ~400 km. To analyse ground motion, all macroseismic data were requested from the above mentioned institutes (Table 2). Only the IDP's from the BGS were not provided.

**TABLE 2**

**FIGURE 9**

### 4.4.1. BGS

1860 felt reports were submitted to the BGS, most of them within a radius of 75 km from the epicentre (BGS, 2015a). People were awoken from their sleep because of a moderate shaking effect and, sometimes, a well perceived sound. Most responses came from the cities of Ramsgate and Margate and from their hamlets (BGS,



2015a). The event was also well perceived in Dover and Folkestone, respectively 21 km and 31 km SW of the epicentre, and in Hernebay (NW, 23 km). The furthest location to which an intensity could be assigned was Norwich (N, 145 km, intensity II). The most distant reports have been received from near Hunstanton (195 km NW), and also some sole reports from Leicester, Milton Keynes and Andover. Remarkably, although London (~100 km, W) is at closer distance than Norwich, only a few responses were submitted from the capital. Apparently, big noise cities do far worse in reporting earthquake perception than smaller towns. IDP's were not available from the BGS but the map of felt locations was updated in real-time on the BGS website (BGS, 2015b). The BGS groups IDP's from their online survey in 25 km$^2$ cells using postcodes (and thus no geocoded addresses). Each grid cell is treated as a locality, reports are grouped accordingly and an intensity (EMS-98) is assigned to each locality from the aggregated data, given at least five responses were submitted from in that cell (BGS, 2015a). For cells with less than five answers, the location is indicated as *'Felt'* and no intensity is assigned. To 58 cells an intensity could be assigned (Fig. 9). The grid cell intensity distribution is dominated by intensities IV and V (Fig. 9) because of the majority of near-field reports. An intensity of $I_0 = V$ was assigned to the epicentre area. This value, however, may be underestimated as, on Twitter, Mr. H. Smith reported cracks in his house and showed a few broken fallen roof tiles at Flete (Kent county, 8 km from the epicentre). To explore potential damage degree 1 or 2 in vulnerability classes A/B in the epicentral area, which is difficult to obtain from internet forms alone, a macroseismic field enquiry (cf. Sira 2015) should have been performed. The epicentral internet intensity of the French 2016 La Rochelle earthquake, for example, gave an $I_0 = V$ whereas the post-seismic field survey resulted in maximum intensity VI (Sira et al. 2016). The cell at Flete near Ramsgate map may be tentatively increased up to intensity VI, however, this has to be confirmed in the field. No other damage has been reported elsewhere.

### 4.4.2 BCSF

The BSCF received 250 online individual witness reports and 135 official community reports after sending out a request through various online and official formularies in France. To 83 of the 168 localities that responded, an intensity could be assigned. Between 45 km and 80 km epicentral distance, the event was well perceived and a maximum intensity of IV could be assigned to 14 localities in the departments of Pas-du-Calais and du Nord (Sira and Schaming, 2015). These are all localities at the French coast of the North Sea. Intensity III could be observed in the cities of Calais and Boulogne-Sur-Mer. Some rare and rather uncertain intensity III values from more than 100 km probably result from the low amount of responses (< 3) within these localities. Following the EMS-98 guidelines: i.e. *the tremor is only felt at isolated places (> 1%) of individuals at rest and in specially receptive position indoors*, these higher intensities were rescaled to intensity II.

### 4.4.3 ROB-BNS

The ROB-BNS received 1980 individual responses from people living in Belgium, six entries from France, six from The Netherlands and one from the UK. After removing false entries, geocoding and floor-level correction, 1617 intensities remained. Many reports were submitted from cities at the Belgian coast (between 85 km and 120 km epicentral distance) and from the provinces of West-Flanders, East-Flanders and Hainaut. After following the grid cell methodology, the Ramsgate intensity distribution shows a WNW-ESE orientation (Fig. 10). Contrary to the French coast, intensity IV was nowhere reached. Intensity III was assigned to the whole Belgian coast and to almost entire western Belgium, including cities of Bruges (III, 125 km), Ghent (III, 160 km) and Brussels (II-III,



210 km). South of Brussels, intensity III was sometimes reached in isolated places. North of Brussels, intensity II alternates with areas with intensity I (not felt) or areas without enough responses to assign an intensity to a grid cell, although some sole reports mostly indicate intensity I. Reports from large distances lead to intensity assessment of populated cities, such as Antwerp (II, 205 km), Liège (II, 300 km) and Verviers (II, 320 km).

5 Individual intensity I reports indicate that the event was not perceived in the NE part of Belgium.

### 4.4.4 EMSC

1595 thumbnails and 368 questionnaires (divided in clusters; see 4.3.4) were submitted to the EMSC. To 39 clusters a reliable intensity could be assigned. The EMSC gathered a substantial amount of data, yet almost only from the UK as clusters are almost all located near the epicentre and intensity VI and V are most frequent (Fig.

9). The thumbnail distribution is more scattered as more reports from larger epicentral distance were submitted. This lead to a normal distribution with mostly intensity III and IV. Only a minor amount of 'not felt' reports were submitted. The Ramsgate event was perceived until Grimsby and Stoke-On-Trent, respectively 270 km NNW and 310 NW from the epicentre. However, these sole reports probably concern reports from people living at higher floors which is not taking account in attribution of the intensity value.

### 4.4.5 USGS

A rather limited amount of responses were submitted to the USGS after the Ramsgate earthquake. With 156 individual responses, an intensity could be assigned to 14 grid cells of 100 km$^2$. A maximal intensity of IV was assigned to the epicentral area.

### 4.4.6 Merging databases

**FIGURE 10**

With respect to the epicentre, the 2015 Ramsgate earthquake was felt from Liège and Verviers (BE) in the ESE, Lille and Calais (FR) in the SE, Norwich (UK) in the North, Stoke-On-Trent in the NE and Brighton in the SW. For the four countries involved, **6,222** people responded to the online questionnaires of five different institutes to have felt the Ramsgate earthquake (Table 2). Similar as to the Goch earthquake, the complex macroseismic

situation in Europe is reflected in the fragmented datasets of the Ramsgate earthquake, but the subtotal of entries to national institutes (Fig. 9) fully covers the felt area. To illustrate the shaking effect homogeneously, a 100 km$^2$ grid cell intensity map was created after merging the clustered data of the EMSC, the 25 km$^2$ grid cells of the BGS and after calculating grid cell intensities for the USGS, BCSF and ROB-BNS IDP's (Fig. 10). The intensity distribution is clearly WNW-ESE oriented. Intensity VI was assigned to one cell. Intensity V was reached within

35 km from the epicentre. Intensity IV occurs in the whole Kent county, at isolated locations 75 km north of the epicentre and at the French coast. Intensity III occurs up to 150 km in the UK but up to 275 km on the continent. Intensity II was reported at some isolated locations in the UK, in the area north of Brussels in Belgium and around Arras in France. Intensity I is almost everywhere lacking except to a limit extend in the NE of Belgium.

### 4.4.7 Attenuation analysis

**FIGURE 11**





The Goch intensity assessment showed that deriving an intensity attenuation relation (IAR) through grid cell intensities leads to an equal-area IAR that is not biased by concentration of individual responses due to population density. This result offers a great perspective in sharing macroseismic data between seismological agencies that all have their own inquiry and their own way of clustering data because intensity cells can be exchanged without
sharing any personal information. We test this method on the Ramsgate earthquake: an attenuation model is derived through the centroid data from the 25 km$^2$ cells of the BGS, the clustered data of the EMSC and 100 km$^2$ intensity cells generated from ROB-BNS, USGS and BCSF IDP's. The Ramsgate IAR is compared to a CEUS prediction model for an $M_W$ 3.7 event (equivalent of an $M_L$ 4.2 event) at 15 km depth. We only derived an IAR through grid cell intensities as an IAR derived from individual IDP's would fail in the epicentral area because
IDP's from the BGS were not available. In the distance histogram (Fig. 11), the highest amount of cells is present in the epicentral area because different grid sizes are used. The epicentral intensity of the derived IAR ($I_0$ = 4.5) is slightly lower than the I = V indicated by the BGS grid cells (and the I = IV at Flete) because the IAR is calculated from the mean value of grid cell intensities in the first 20 km. Intensity attenuation occurs in the first 100 km at a lower rate than predicted by the CEUS model. This is probably related to the underreporting of low
intensities in the first 100 km because the BGS *Felt* intensities (white cells in Fig. 10), which probably corresponds to II-III, cannot be included in the attenuation analysis. This drives IAR to intensity overestimation in the near-field. A change in attenuation rate is present at 90 km. Relating this bump to a Moho bounce effect remains subjective: Ramsgate was a coastal event and the IAR suffers from a lack of responses from (i) the North Sea area and also (ii) from the London area from where mostly only *Felt* reports are available. Interestingly, the Ramsgate
earthquake was much further felt and reported on the continent than in the UK. Between 100 km and 300 km, the attenuation model is almost fully derived from intensities with an azimuth ranging between 90° to 120° (Fig. 11). At these distances, the smoothed mean attenuation of the continent follows the CEUS model consistently.

## 5 Discussion

Depending on thickness and seismic impedance of surface sediments in a basin, surface ground motions can be
strongly amplified and deamplified as well, resulting in large differences in site response. Prolonged scattering of long-period waves in a sedimentary basin is the critical factor for surface ground motion amplification. Intensity amplification in sedimentary basins, near-field as well as at large epicentral distances, have been the subject of site response studies using solely macroseismic data (e.g. García Moreno and Camelbeeck, 2013; Haak et al., 1994; Martin et al., 2015; Sbarra et al., 2012a). The Goch and Ramsgate macroseismic data show significant
deviations from the simple model of concentric isoseismals. The perception area of the Goch earthquake is stretched towards SW but lacked any responses at closer distances in that direction (NE Belgium). The Ramsgate event has a pronounced WNW-ESE distribution pattern but also lacks any observations at large distances in azimuthal orientations other than 90° to 120°. The Goch non-concentric shaking distribution is not a matter of population density as the affected area is densely populated (Fig. 8).

35          We also computed the P- and S-wave far field radiation pattern (Aki & Richards, 2002, Eq 4.29) of the Goch strike-slip focal mechanism. The P-wave energy is largest in a SW direction while the S-wave energy is largest in the S and E directions. If P-waves were responsible for the felt reports, then a majority of people in the NE Belgium should have reported the earthquake as the whole area falls inside the highest displacements lobe at the source. If S-waves are responsible for the felt perceptibility, which is mostly the case for the Goch earthquake



(S-waves have higher amplitudes than P-waves), only the Namur-Hasselt-Maastricht line falls in the lower radiated amplitudes. Although the S-wave radiation towards Antwerp is larger than towards Brussels, with the former at closer distance than the latter, yet, only three reports were submitted from Antwerp versus 83 from Brussels. Hence, the lack of responses in NE Belgium cannot be linked to the source mechanism.

5        To explain the observed intensity patterns, we thus need to address the effect of regional geology, especially bedrock depth, on seismic wave attenuation of frequencies to which humans are susceptible, both in the LRG and in and above the A-BM. Figure 12 shows the combined shaking effect of Ramsgate and Goch, illustrated on a map of the depth to seismogenic bedrock. The top of the seismogenic bedrock is here interpreted as the surface with highest acoustic impedance contrast between soft sediments and basement. In central and

northern Belgium, the seismogenic bedrock either corresponds to the top of the A-BM or, when present, the top of the Cretaceous. The A-BM is present at the surface in incised river valleys in central Belgium and its top gradually deepens towards the Belgian-Dutch border where it is covered by 800 m of soft sediments (Legrand, 1968). In The Netherlands, the top of bedrock coincides with the base of the Lower North Sea Group (Van Adrichem Boogaert and Kouwe, 1993-1997) and reaches a maximum depth of 1500 m in the centre of the LRG

below Eindhoven and 's Hertogenbosch (NL). In Germany, the seismogenic bedrock corresponds to the base of the Tertiary sediments in the LRG. Its deepest part (> 1300 m) is located SW of the Erft / Swist Fault system in the southern part of the LRG.

**5.1 Ground motion (de-)amplification due to sediment thickness**

**FIGURE 12**

Figure 12 suggests that the felt-area of the Ramsgate and Goch earthquakes, apart from geometrical amplitude attenuation, is controlled by the thickness of the sedimentary cover:

- Although Goch was reported from Cologne and Bonn (DE), no reports came in from SW of the Erft / Swist fault system in the LRG, i.e. the deepest part of the LRG in Germany.

- In NE Belgium macroseismic responses are absent both for Goch and Ramsgate. For Goch no responses
are submitted from places at large bedrock depth, but the event is reported between Liège and Brussels where bedrock is at rather shallow depth ($\leq 100$ m).

- Integration of bedrock depth and the WNW-ESE oriented Ramsgate intensities on the continent shows that the farther from the source, the thinner the sedimentary cover above the A-BM must be in order to have perceived the event (Fig. 13). Regression analysis moreover indicates that the relation between distance
and depth gets steeper for lower intensities.

**FIGURE 13**

Based on these observations it can be concluded that at a close distance to the Goch earthquake source (< 60 km), seismic energy was capable to penetrate the thick sedimentary cover in the LRG causing the earthquake to be felt at the surface. At larger distance, however, the thick sedimentary cover in NE Belgium and in the LRG behaved

as a low-pass filter and attenuated seismic energy at higher frequencies to which human and 1-3 story buildings are susceptible. In central Belgium, moreover, the E-W elongated shape of the felt shaking was not due to a better transfer of seismic waves in the E-W direction, but is related to an E-W elongated zone where shallow bedrock depth ($\leq 100$ m) amplified seismic waves in the susceptible frequency range. This conclusion is corroborated by intensity increase in the IAR (Fig. 7B) and indicates ground motion amplification at Brussels.





For similar-size magnitude earthquakes, shallow events are more focused, have higher epicentral intensities and are less far felt than deeper events (Kövesligethy, 1907). Indeed, the 15 km deep Ramsgate earthquake was felt much further (~400 km) than the 10 km deep Goch event (~180 km). In contrast, the epicentral intensity ($I_0 = 3.7$) of the Goch earthquake is lower than any prediction model and is considerably lower than the epicentral intensity ($I_0 > 5.0$) of the Ramsgate earthquake. Such a low epicentral intensity of 3.7 would indicate a source of 25 km deep, which contrasts with the instrumental 10 km. Hence, the low epicentral intensity can only be explained by a local site effect, i.e. the sedimentary cover thickness (~600 m at Goch; Fig. 12), attenuating higher frequencies at the epicentre. A similar effect has been reported from the 17 km deep, $M_L$ 5.8 1992 Roermond (NL) earthquake (Camelbeeck and van Eck, 1994) where epicentral intensity was lower than predicted and did not exceed VII on the MSK-scale (Haak et al., 1994; Meidow and Ahorner, 1994). Also this was explained due to significant absorption of energy by 1500 m of sediments below Roermond (Ewald et al., 2006).

The Goch grid cell map shows higher intensities on the Venlo Horst (shallow bedrock), NE of the Goch epicentre, and lower epicentral intensities and thus site effect attenuation in the Venlo Graben, SW of the epicentre, due to the deeper bedrock. This observation supports the conclusion that in a NW-SE elongated area sediment thickness strongly influenced ground motion of the Goch earthquake.

East of the epicentre, NRW-GD received many reports from the Münster Basin (Fig. 12): i.e. the asymmetric synclinal basin which basement is made up of Palaeozoic and pre-Cretaceous Mesozoic rock and which is predominantly filled with soft marls up to 1400 m. In its centre, the seismogenic bedrock is considered to be at 2000 m depth (Bilgili et al., 2009; Richwien et al., 1963). At Münster, the Goch earthquake resulted in a higher grid cell intensity (Fig. 7) which might be related to a basin effect. The large-distance responses reported from the southern border of the Münster Basin (up to 170 km) can be related to basin-edge effects.

## 5.2 Q factor(s)

Although it sounds contra intuitive that a deep basement causes damping instead of long period amplification, these M4 earthquakes have to be seen in perspective with respect to the depth and distance from the source. In the past, larger-magnitude earthquakes have generated ground motions that were felt in the non-felt region of Goch and Ramsgate earthquakes. The 2002 $M_L$ 4.9 Alsdorf earthquake ($I_0 = $ VI) (Hinzen, 2005), for instance, which occurred at the SW borders of the LRG, was widely perceived across the whole northern Belgium, in the southern Netherlands and in W Germany. Also the 1938 Zulzeke-Nukerke earthquake ($M_S$ 5.0; $I_0 = $ VII; see Fig. 3 for location) was felt in NE Belgium at places with the largest bedrock depth. Damage reports of this event showed an E-W oriented, elongated, non-circular distribution (Nguyen et al., 2004; Somville, 1939, Camelbeeck et al., 2014). Also the largest magnitude earthquake events in the $M_L$ -0.7 to $M_L$ 3.2, 2008-2010 Walloon Brabant seismic swarm (Van Noten et al., 2015a, b) resulted in a predominant E-W macroseismic distribution illustrating the importance of the shallow Brabant Massif on the perception of these events.

The fact that Ramsgate was felt far on the continent is, apart from its larger source depth, related to the shallow Brabant Massif and its thin sedimentary cover in Central Belgium. The anisotropy in the felt distributions of the 2015 Ramsgate and the 1938 Zulzeke-Nukerke earthquakes results from a combination of different Q-factors: Q is very high in the A-BM crust and is low in the sedimentary cover. Hence, at places with a thin cover along-strike (WNW-ESE) of the A-BM, seismic waves are only attenuated by distance attenuation. Perpendicular to the A-BM tectonic axis, however, ground motions are attenuated (i) to the north by the increasing sedimentary





cover (lower combined Q factor), and (ii) to the south, because seismic waves need to propagate through the WNW-ESE oriented, crustal rooted faults of the Kent-Artois Shear Zone (Fig. 12), which has a lower Q due to its fractured nature. This explains why (i) why Central Belgium is affected to a higher hazard than northern Belgium and (ii) why hardly any felt reports were submitted south of this shear zone in France. Compared to the wide-

spread effect on the continent, the felt effect of the 2015 Ramsgate earthquake is rather limited in the UK as hardly any reactions were submitted from > 150 km distance. Following a similar reasoning as on the continent, this less wide-spread effect might be related to attenuating basin effects in the Great London Basin and to intensity attenuation across the Kent-Artois Shear Zone.

## 6 Conclusions

Construction of transfrontier macroseismic maps in Europe remains challenging due to a variety of questionnaires and differences in intensity calculation (up to one intensity unity) between different seismological agencies. These differences highlight the importance of the European macroseismic scale for merging data. It is necessary to collect the highest possible amount of questionnaires to ameliorate the liability of the macroseismic intensity analysis and national agencies are in the best place for that as they still gather the highest number of responses

(see subtotals in Figs. 5 and 9). However, harmonisation between institutional questionnaires is necessary in order to use the full amount of transfrontier collected testimonies.

We demonstrated the functionality of geocoding and applying the grid cell method for merging (inter)national macroseismic databases. Although less data points are used, geocoding intensity locations minimizes the location uncertainties in a macroseismic map. A grid cell intensity map shows a more conclusive

intensity distribution than a ZIP code intensity map. Grid cell maps are self-explanatory and avoid errors on the often subjective contouring for isoseismals. Gridding moreover allows merging multiple DYFI macroseismic datasets collected by different seismological institutes in which the individual locations are (made) imprecise. Sharing grid cell information between seismological institutes can thus be a solution in macroseismic data exchange of transfrontier-felt earthquakes because all statistical intensity information is stored in the grid cells

and legal issues, such sharing a person's privacy, are avoided.

Post-processing internet macroseismic data strongly improves the quality of real-time intensity evaluation of individual agencies. For a reliable post-seismic transfrontier intensity assessment, we advise to start with the inquiry of the institute that has the best spatial cover. Geocode if possible (if using Google remind the 2500 address limit/day) and make sure a floor-level correction can be applied; if not, add the question to your

enquiry to avoid intensity overestimation. Create a network of volunteers that can be alerted in real-time to obtain more intensity data and increase the number of not-felt, intensity I entries. Collect and merge other institutional data to obtain the best coverage of the felt effect.

Macroseismic grid cell intensity analysis of the $M_L$ 4.3 2011 Goch and the $M_L$ 4.2 2015 Ramsgate earthquakes proved that the non-concentric macroseismic distributions of both events are controlled by distance

attenuation, the Q-factor in the bedrock and in the overlying sedimentary column, and by bedrock depth. For both earthquakes, a regional amplification effect occurs in central Belgium at shallow sites, even at places with a considerable distance to the sources. In NE Belgium and in SW Lower Rhine Graben, absence of macroseismic responses is related to seismic energy absorption by the thick sedimentary cover. The Ramsgate earthquake was farther felt than the similar-size magnitude Goch earthquake because of its larger source depth but predominantly





due to an efficient wave propagation and low attenuation along the WNW-ESE tectonic axis of the Anglo-Brabant Massif, enhanced by the focal mechanism generating a directivity effect. These results emphasize the importance of an ancient tectonic structure on anisotropic wave propagation: seismic waves are less attenuated within a seismotectonic unit but are stronger attenuated along travel paths perpendicular to the structural axis of the

basement massif, particularly when they propagate through adjacent rheologically different units that are separated by deep crustal-rooted faults or that are buried by a sedimentary cover.

**Data Availability**

Topography data in Fig. 3 are from Shuttle Radar Topography Mission (SRTM) available from the U.S. Geological Survey. All macroseismic data were requested from the corresponding seismological institutes (See

Appendix A). Bedrock data in Figure 12 are openly available from DOV (https://dov.vlaanderen.be) and DINOLOKET (https://www.dinoloket.nl/). All other data and maps used in this paper came from published sources listed in the references. Shapefiles to regenerate Figures 6 and 10 are available in Supplementary material.

**Acknowledgements**

The authors sincerely thank the 16,503 individuals that responded to both earthquakes via different online

platforms. We further acknowledge the following people for providing the institutional macroseismic data: B. Dost and G.-J. van den Hazel (KNMI), K. Lehmann (North Rhine-Westphalia Geological Survey), R. Bossu and G. Mazet-Roux (EMSC), D. Wald and V. Quitoriano (USGS), and B. Baptie and S. Sargeant (BGS). R. Gold and the Seismology.be team are thanked for discussions on the content. Figures were created in QGIS (2016). K. Van Noten is supported by the Fonds de la Recherche Scientifique (FNRS - Belgium) under grant PDR T.0116.14.

**Declaration of interest**

The authors declare that they have no conflict of interest.

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

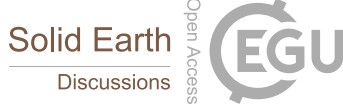



**Table Captions**

**Table 1: Institutional macroseismic data for the 2011 Goch earthquake. When street addresses were available, response locations were geocoded for location improvement and intensities were floor-level corrected. Only total amount of geocoded and corrected intensity points are used for final intensity assessment in this study.**

| Event | Agency | ML | MW | Nr of Inq | Geocoded & corrected | depth | I max (EMS-98) |
|---|---|---|---|---|---|---|---|
| **Goch** | ROB-BNS | 4.3 | 3.7 | 3294 | 2134 | 10 | VI |
| Thursday | KNMI | 4.5 | - | 4222 | 3637 | 10.3 | VI |
| 8/09/2011 | NRW-GD | 4.3 | - | 1199 | 1149 | 11 | V |
| 19:02:50 UTC | **National agencies subtotal:** | | | **8715** | **6920** | | |
| | EMSC | 4.3 | - | 528 | 259 | 10 | V |
| | USGS | - | - | 1038 | 1038 | - | VI |
| | **Total Responses:** | | | **10281** | **8217** | | |

**Table 2: Institutional macroseismic datasets for the 2015 Ramsgate earthquake. When street addresses were available, response locations were geocoded for location improvement and intensities were floor-level corrected. Only total amount of geocoded and corrected intensity points are used for final intensity assessment in this study.**

| Event | Agency | ML | MW | Nr of Inq | Geocoded & corrected | depth | I max (EMS-98) |
|---|---|---|---|---|---|---|---|
| **Ramsgate** | BGS | 4.2 | | *1860* | 0 | 12 | V (epi) |
| Friday | BCSF | 4.0 | | 250 | 216 | 10 | IV |
| 22/05/2011 | ROB-BNS | 4.1 | | 1993 | 1618 | 15 | III |
| 01:52:17 UTC | **National agencies subtotal:** | | | **4103** | **1834** | | |
| | EMSC | 4.2 | | 1963 | 1595 | 10 | V (epi) |
| | USGS | 3.7 | | 156 | 156 | 10 | IV |
| | **Total responses:** | | | **6222** | **3585** | | |

**Figure Captions**

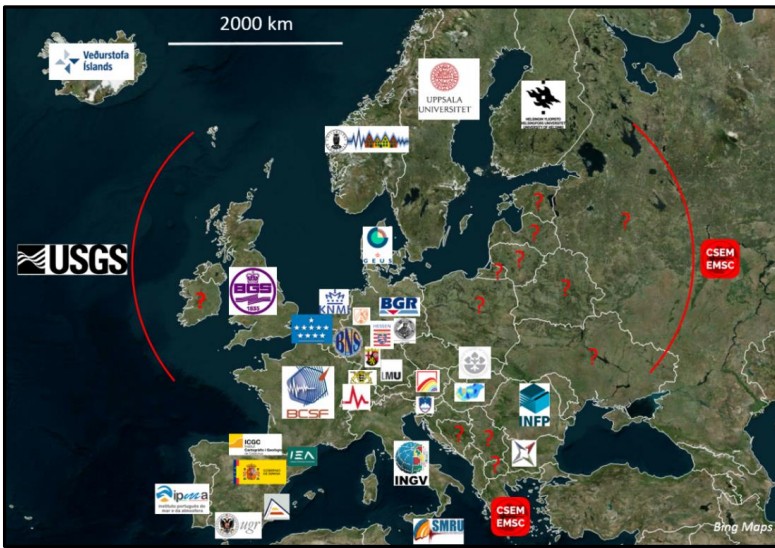

**Figure 1: Illustration of 31 European seismological institutes and 2 international (EMSC and USGS) that provide online *"Did You Feel It?"* inquiries over the European territory. See Appendix A for institutional names and hyperlinks to the online questionnaires. The question mark indicates countries for which an online enquiry is absent or has not been found.**





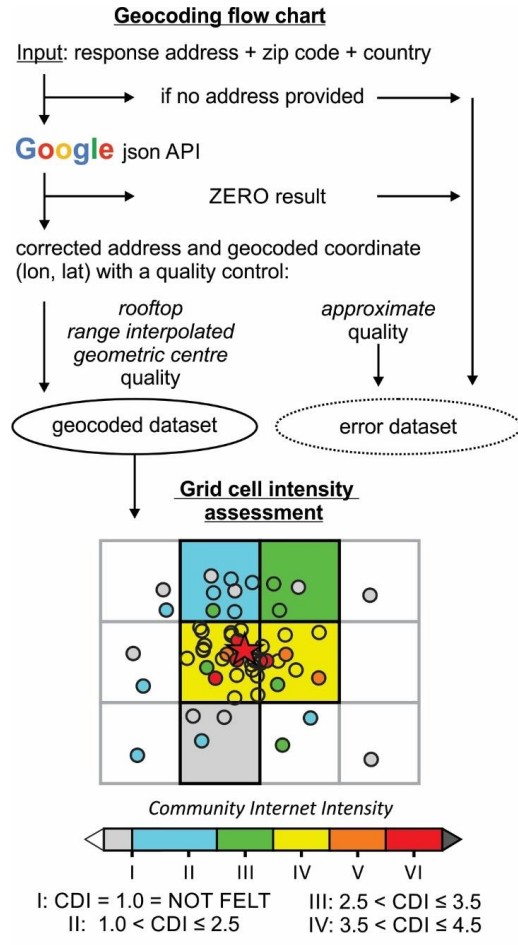

**Figure 2: Geocoding flow-chart and grid cell intensity procedure. Input addresses are send to a Google™ Json API in which addresses are corrected and geocoded. Precision of the geocoded coordinates is quantified by a quality factor. For absent, unrecognised or unprecise (too low quality) addresses, the geocoded response is not used for grid cell intensity assessment. All geocoded addresses are grouped within 100 km² grid cells. Cells with at least three responses are coloured to the average intensity.**





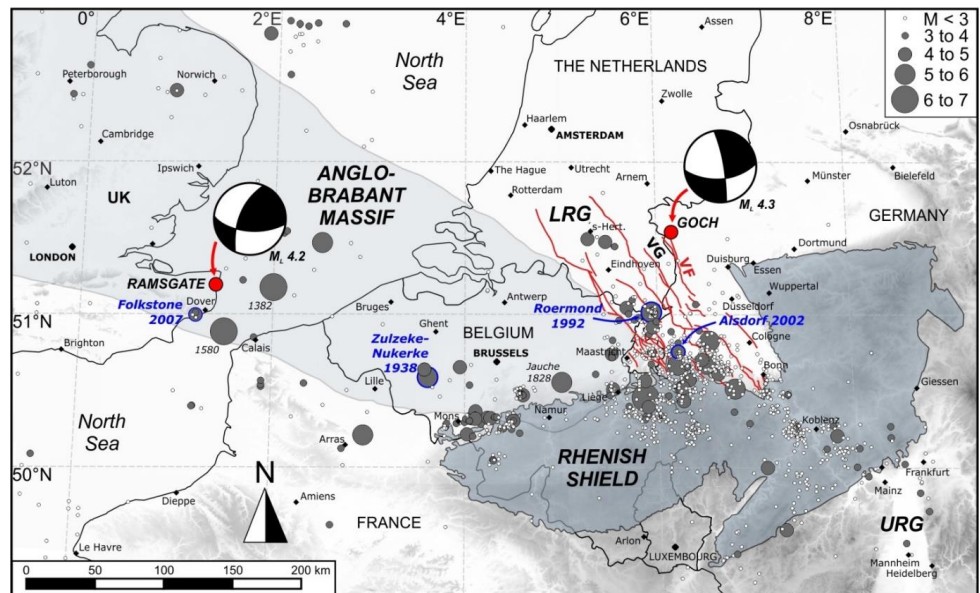

**Figure 3: Combined ROB-BNS and BGS instrumental and historical seismic catalogues illustrated on the 30 m SRTM-DEM. The 2011 $M_L$ 4.3 Goch earthquake occurred at the NE end of the Venlo Graben (VB), close to the Viersen Fault (VF) in the NE part of the Lower Rhine Graben (LRG). The 2015 $M_L$ 4.2 Ramsgate earthquake took place off-shore**

5    **UK within the basement rocks of the Anglo-Brabant Massif which extends from northern Belgium to SE UK. URG: Upper Rhine Graben. Macroseismic surveys of earthquakes indicated in blue are discussed in the text.**





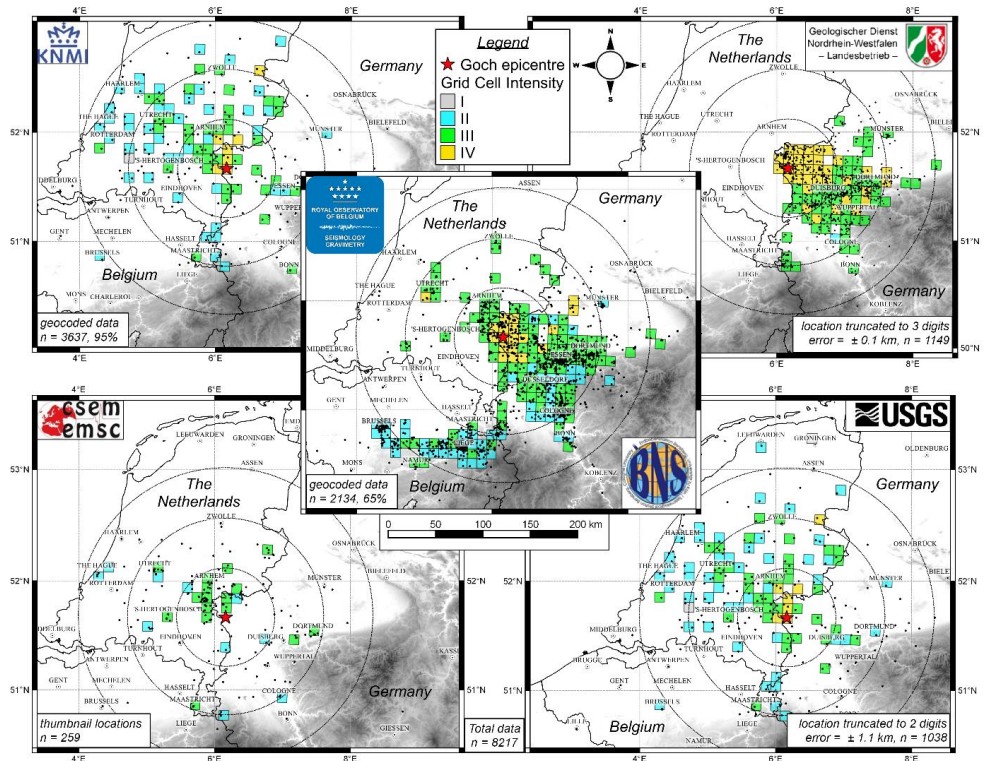

**Figure 4: Fragmentation of macroseismic data of the 2011 Goch earthquake. Transfrontier collaborations (ROB-BNS) and international institutes (USGS, EMSC) that provide an English or a multilanguage inquiry cover a wider geographical spread than national inquiries (KNMI, NRW-GD). ROB-BNS and KNMI datasets are geocoded. Grid cell size is 100 km². Only cells with at least three entries are coloured. Total responses = 8217. Rings represent 50 km, 100 km and 150 km epicentral distance.**





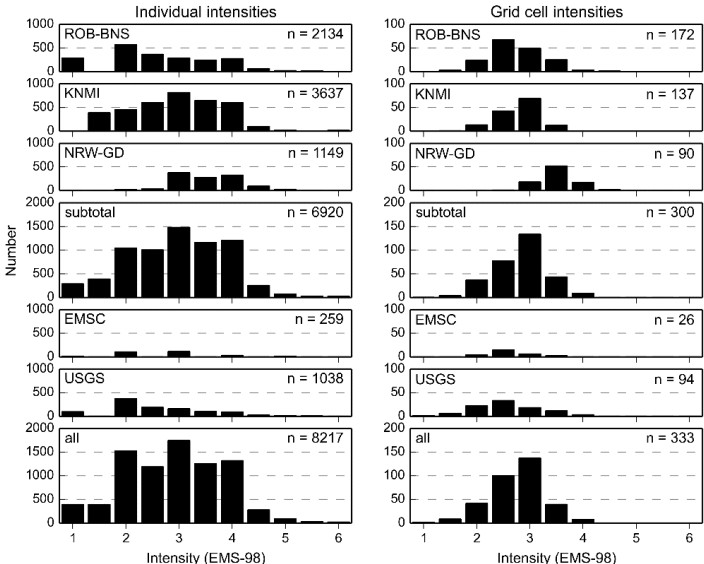

**Figure 5: Goch earthquake intensity distributions binned per half intensity units. (Left) Intensity distribution of the (geocoded, if possible) IDP's. NRW-GD lacks intensity I or 'not felt' entries. (Right) Grid cell intensity distributions. Note the appearance of a normal distribution after grid cell calculation. The 'subtotal' distribution (total number of responses submitted to 'national' agencies) is considerably larger than the EMSC and USGS distributions.**

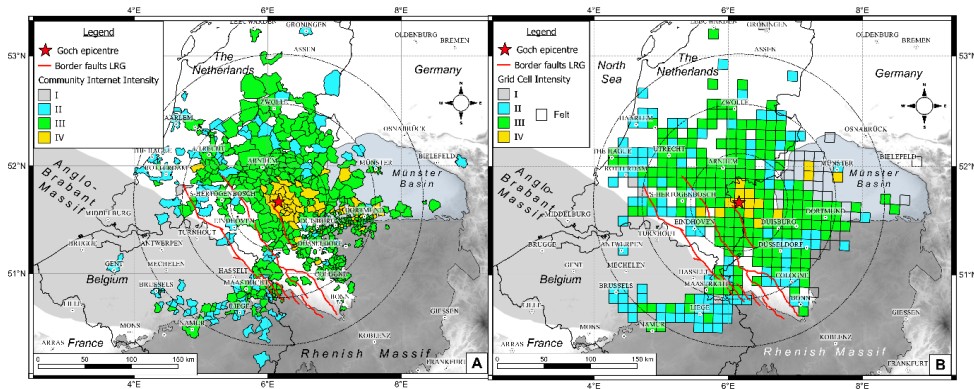

**Figure 6: 2011 Goch earthquake intensity maps. A) ZIP-CIIM using all 8217 entries. B) Merged grid cell intensity map (without NRW-GD data; n = 7068) generated with geocoded responses from ROB-BNS, KNMI and intensity data from EMSC and USGS. Note the absence of macroseismic entries in NE Belgium. Grid cell size is 100 km². Rings represent 50 km, 100 km and 150 km epicentral distance.**





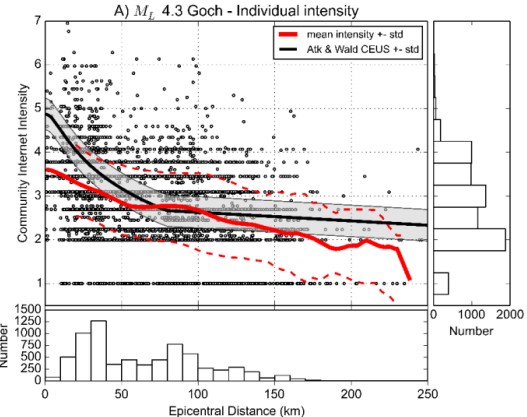

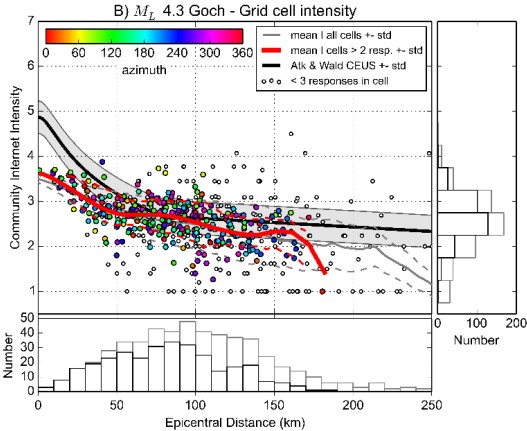

**Figure 7: Intensity attenuation of the 2011 Goch earthquake. The intensity attenuation relation (IAR) is derived from (A) individual IDP's and (B) grid cell centroid intensity data after merging macroseismic datasets of all institutes (except NRW-GD). The smoothed attenuation curves are derived with a moving bin technique, bin window length = 20 km, overlapping steps of 2 km. Red/grey lines = IAR's deduced from cells with ≥ 3 and < 2 responses, respectively. The horizontal and vertical histograms show the number of intensity distance points binned per 10 km and per half intensity unit, respectively. Colours represent the azimuthal position of the grid cell centre (centroid) relative to the seismic source. Black line indicates CEUS predicted intensities using Atkinson and Wald (2007) for an $M_W$ 3.7.**



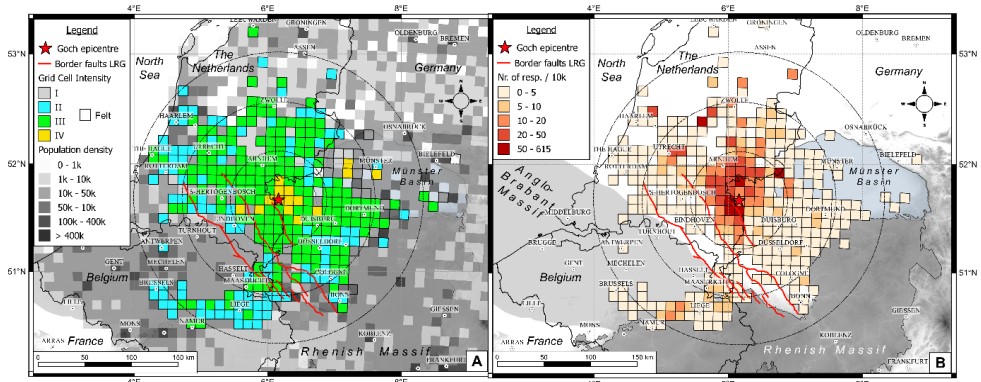

**Figure 8: A) Merged grid cell intensity map overlain on the population density grid cell map. Note the absence of macroseismic entries in the northern part of Belgium, despite its high population density. B) Number of responses normalised for population density (nr / 10k people). On average the Goch event resulted in 9.8 resp./10k inhabitants, with ~40 resp./10k in the epicentral area. Rings represent 50 km, 100 km and 150 km epicentral distance.**

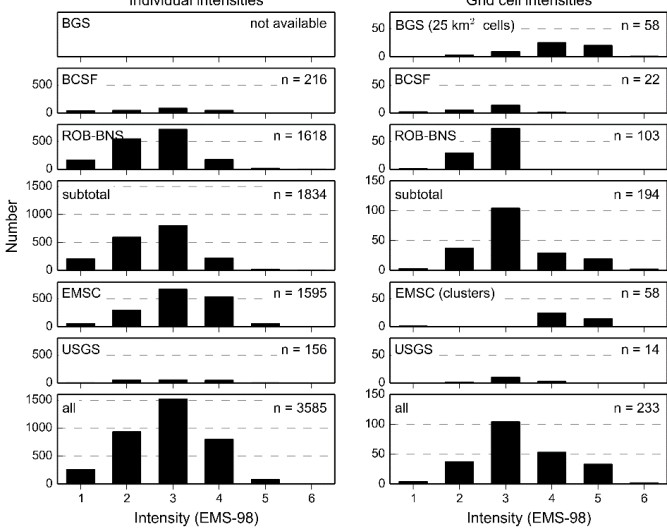

**Figure 9: Ramsgate earthquake intensity distributions binned per full intensity unit. (Left) Individual IDP's. (Right) Grid cells intensities. BGS individual intensities were not available.**





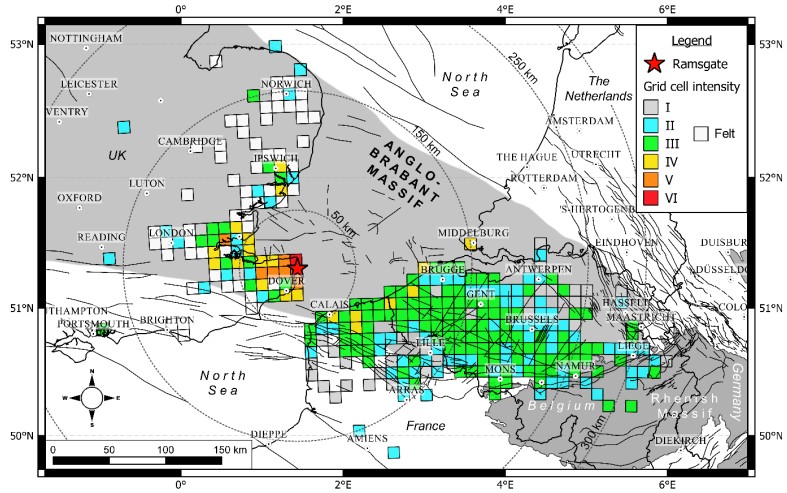

**Figure 10: Grid cell intensity map of the 2015 Ramsgate earthquake. UK grid cells are merged from BGS, EMSC and USGS data. On the continent (merged data from ROB-BNS and BCSF), only cells with at least three entries are coloured. Note the particular WNW-ESE non-concentric distribution following the structure of the Anglo-Brabant Massif. Grid cell size is 100 km².**

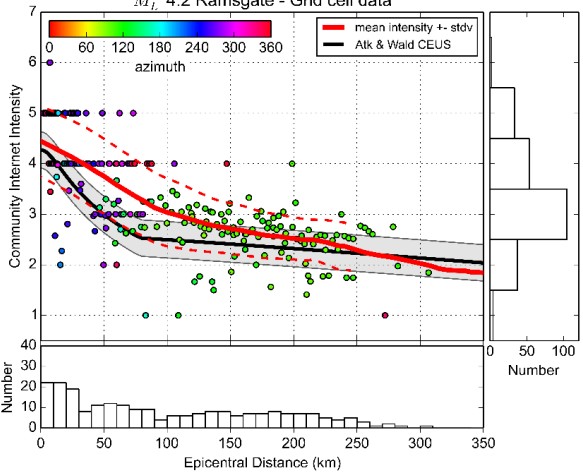

**Figure 11: Intensity attenuation of the 2015 Ramsgate earthquake. The smoothed IAR (red line) is derived from grid cell intensity data from BGS, ROB-BNS, USGS, EMSC and BCSF with a moving bin technique, bin window length = 20 km, moving steps = 5 km. Horizontal and vertical histograms show the number of intensity distance points binned per 10 km and per intensity unit, respectively. Colours represent the azimuthal position of grid cell centre (centroid) relative to the seismic source. Black line indicates CEUS predicted intensities using Atkinson and Wald (2007) for an Mw 3.7.**



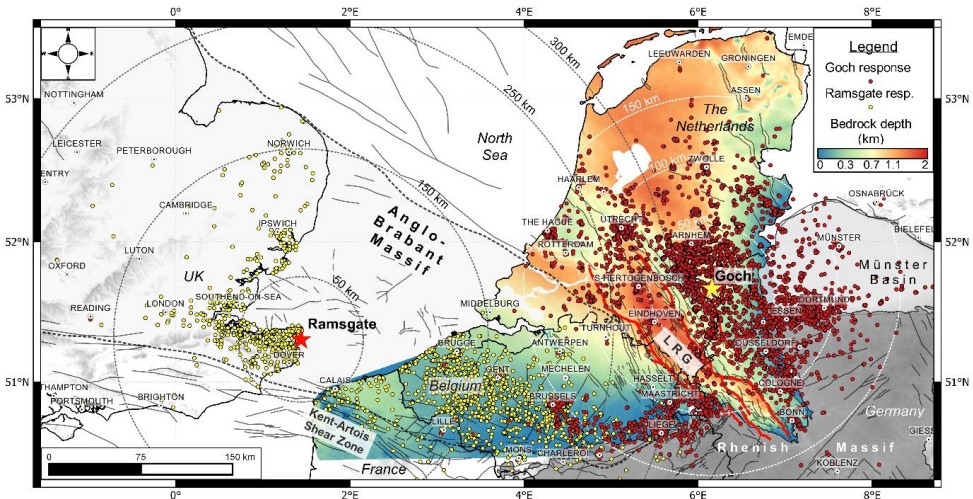

**Figure 12: Combined distributions of the 2011 Goch earthquake (red) and 2015 Ramsgate earthquake (yellow) illustrated on a depth to bedrock map of northern Belgium, The Netherlands and western Germany. Note the absence of felt responses in NE Belgium (between Antwerp and Hasselt), SW of Eindhoven (NL) and SW of Cologne due to thick sedimentary cover. Map compiled from numerical data of the Databank Ondergrond Vlaanderen (2016; Belgian Data) and TNO (Dinoloket, 2016; Dutch data) combined with maps of Legrand (1968), Weber (2007) and Hager and Prüfert (1988). Bedrock depth in the UK is not shown.**

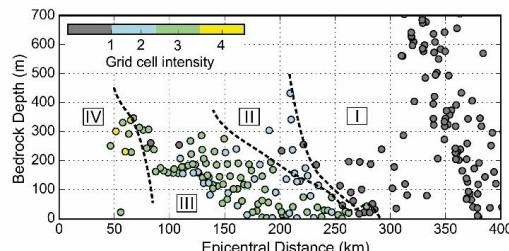

**Figure 13: Bedrock depth of centroids from the continent (see Fig. 12) versus epicentral distance to the Ramsgate $M_L$ 4.2 earthquake. Dashed lines limit maximal depth-distance relation for the indicated intensities.**

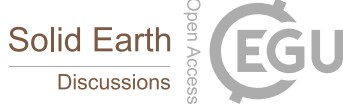

**Appendices**

**Appendix A: Overview of European Seismological Institutes that provide an online "*Did You Feel the Earthquake?*" inquiry. URL links are provided. RT? (Real time?) questions if the gathered data are mapped and illustrated online in real-time: y(es), n(o). Institutional abbreviations are used in the text. See Appendix B for language abbreviations.**

| Country | Institute | Lang | URL | RT ? |
|---|---|---|---|---|
| Andorra | Centre d'estudis de la neu/de la muntanya d'Andorra | cat | CENMA | n |
| Austria | Zentralanstalt für Meteorologie und Geodynamik | ger | ZAMG | n |
| Belgium | Royal Observatory of Belgium and Erdbebenstation Bensberg | nld | ROB-BNS | y |
| | | fra | ROB-BNS | y |
| | | ger | ROB-BNS | y |
| | | eng | ROB-BNS | y |
| Bulgary | National Institute in Geophysics, Geodesy and Geography | bul | NIGG | n |
| Denmark | Geological Survey of Denmark and Greenland | dan | GEUS | n |
| Finland | University of Helsinki | fin | UH | n |
| | | swe | UH | n |
| France | French Central Seismological Office | fra | BCSF | y |
| Germany | Universität Jena | ger | FSU | n |
| | Ludwig-Maximilians-Universität München | ger | LMU | n |
| | Hessisches Landesamt für Umwelt und Geologie (Hessen) | ger | HLNUG | n |
| | Bundesanstalt für Geowissenschaften und Rohstoffe (Hannover) | ger | BGR | n |
| | Landeserdbebendienst (Freiburg) | ger | LGRB | n |
| | Geologischer Dienst Norddrhein-Westfalen (Krefeld) | ger | NRW GD | n |
| | Landesamt für Geologie und Bergbau (Rheinland-Pfalz – Saarland) | ger | LGB | n |
| | Royal Observatory of Belgium and Erdbebenstation Bensberg | ger | ROB-BNS | y |
| Hungary | Hungary Earthquake Information System | hun | HUN-Reng | n |
| Iceland | Icelandic Met Office | isl | IMO | n |
| Italy | Istituto Nazionale di Geofisica e Vulcanologia | ita | INGV | y |
| Malta | Seismic Monitoring & Research Unit, University of Malta | eng | SMRU | y |
| Netherlands | Koninklijk Nederlands Meteorologisch Instituut | nld | KNMI | n |
| Norway | Norwegian National Seismic Network, University of Bergen | nor | NNSN | n |
| Portugal | Instituto Português do mar e da atmosfera | por | IPMA | n |
| Romania | National Institute for Earth Physics | ron | NIEP | n |
| Slovakia | Earth Science Institute of the Slovakian Academy of Sciences | slk | SAS | n |
| Slovenia | Slovenian Environment Agency | slv | ARSO | n |
| Spain | Instituto Geográfica Nacional | spa | IGN | n |
| | Cartographic Institute of Catalonia | spa | IGC | y |
| | | cat | IGC | y |
| | Granada University | spa | UGR | n |
| | University of Alicante | spa | UA | n |
| Switzerland | Swiss Seismological Service | ger | SED | n |
| | | fra | SED | n |
| | | eng | SED | n |
| | | ita | SED | n |
| Sweden | Svenska nationella seismiska nätet, Uppsala Universitet | swe | SNSN | n |
| UK | British Geological Survey | eng | BGS | y |
| Global | European–Mediterranean Seismological Centre | all | EMSC | y |
| Global | United States Geological Survey | eng | USGS | y |





**Appendix B: Abbreviated languages. The "earthquake" translations are used to find internet inquiries online. Abbreviated languages follow the international three-letter alpha-3/ISO 639-2 code.**

| Languages | "earthquake" |
|-----------|--------------|
| bul | ЗЕМЕТРЕСЕНИЕ |
| cat | terratrèmol |
| dan | jordskælv |
| eng | earthquake |
| fin | maanjäristys |
| fra | tremblement de terre |
| ger | Erdbeben |
| hun | földrengési |
| isl | jarðskjálfta |
| ita | terremoto |
| mlt | terremot |
| nld | aardbeving |
| nor | jordskjelv |
| por | sismo |
| ron | simtit |
| slk | zemetrasenie |
| slv | potres |
| spa | terremoto |
| swe | jordskalv |