# Peer review of "Path and site effects deduced from transfrontier internet macroseismic data of two recent M4 earthquakes in NW Europe"

_Solid Earth, 2016_

## Referee Comment (RC1) · R. Bossu (Referee) · 12 Jan 2017

Path and site effects deduced from transfrontier Internet macroseismic data of 2 recent M4 earthquakes in NW Europe Review by R. Bossu

This ambitious paper covers 2 different topics. It proposes a way how to spatially merge different Internet macroseismic data and it proposes an explanation of the obtained macroseismic maps in terms of path and site effects.

It is a rather long article with a rich list of references and is generally well written article. It covers an important topic which is how to merge Internet macroseismic data collected at national level for transfrontier earthquakes (as explained below this issue is not fully

[Figure]

covered and only address the spatial grouping of individual observations). This topic is important because national institutes generally collected many times more data that international organisation such as the USGS or EMSC.

There are however a number of points that could improve the readability of the manuscript and its overall quality • In the text, the authors have been using Âń Did you Feel it Âż not only to present the tool developed and operated by the USGS but also more generally for Âń Internet macroseismic data Âż. (One should note that the title do use the latter). I find this confusing. I believe this distinction should clearly appear in the text. DYFI was the very first online macroseismic tool, several institutes have implemented the same questionnaires, but others have developed their own approach.

• My second issue is about the description of the data used. I believe that a description of the methodology for each data provider (perhaps in appendix) would be useful. Is it a questionnaire (or thumbnails), how are the locations determined (zip code, geocoded full address, nearest city which was an option at EMSC when eyewitnesses declined to provide their full address) and how the intensity is assigned.

• The paper makes a very strong assumption (last sentence of page 2) that intensities may slightly differ from one country to the other (due to differences in questionnaire and/or intensity assignment procedures). Some of the data presented in this paper contradict this statement: the EMSC macroseismic data derived from questionnaires had to be excluded because they differ too much from the other datasets. (For information, these excluded intensities had been assigned by an algorithm developed by one of the father of the EMS98 scale). What I want to stress here is that there is no reference to such a statement and my own experience, or recent by Hough, Martin et al comparing macrosemsic datasets for Ghorka earthquake do not support it. This is probably too much work to fully address this issue, but the assumption that differences in intensity from one country to another are slight should be made clear and explicit.

• A consequence of the previous point is that the methodology is about the spatial grouping of different Internet macroseismic data only.

• In the second paragraph of the introduction, the EMSC is presented alongside the national institute while it works similarly to the USGS Did you feel it. This seems to indicate there is no transfrontier and international internet macroseismic data collection in Europe or that EMSC works at national level, which is not the case.

• There is an incomplete sentence at the beginning of paragraph 2.2

• There are a number of inaccurate statements:

o EMSC does not request not felt response from volunteers. The LastQuake app send notification after felt earthquake to people in the area and some of them may react to this notification by sharing their testimony o First sentence of the conclusion is inaccurate. Transborder macroseismic maps exist in Europe at EMSC. The challenge is to create a denser and possibly more accurate one by merging national datasets. o Third paragraph of the conclusion: the paper does not demonstrate "strongly improves the quality of real time intensity evaluation of individual agencies". Neither intensity assignment nor real time processing is covered in this article

In conclusion, I believe this paper is important. It covers a variety of issues from spatially merging Internet macroseismic datasets to variations of attenuation. There are however a number of shortcomings and inaccuracies to be corrected which will further improve the quality of the paper.

---

## Referee Comment (RC2) · V. de Rubeis (Referee) · 12 Jan 2017

se-2016-150 Submitted on 03 Nov 2016 Path and site effects deduced from transfrontier internet macroseismic data of two recent M4 earthquakes in NW Europe Koen Van Noten, Thomas Lecocq, Christophe Sira, Klaus-G. Hinzen, and Thierry Camelbeeck

The paper deals with macroseismic intensity analysis when an earthquake effects interest more than one country and each one State organization/agency uses its own on-line questionnaire. The paper has several merits: it deals with crowdsourced intensity data collected through online surveys; it attempts to merge different sources intensity data, analyzing factors that operate into data compatibility issues; it raises an interesting criticism to post-code geolocation of intensities, proposing a more physically reliable 10 km square grid geolocation, less critical than administrative driven postcode. An explanation to anisotropies of intensities found of the examined earthquakes is proposed, in terms of Q analysis and geological structures. While I find the paper interesting and potentially publishable, I have to remark some criticism on issues needed to be addressed prior final paper acceptance. Here it follows a detailed list of criticisms to be addressed:

General questions: Did you take into account the percentage of not felt to asses an intensity degree, as macroseismic scale recommends? Can you try to compare attenuation laws for each data sources?

Page 2, line 3: change real time to quasi real time. P2,l11: not complete. Pattern depends on source also, change the sentence like: Concentric pattern of intensity decay is only a theoretical very first approximation, which may serve only to indicate seismic epicentral best location. P2,l25 and l34: explain the meaning of real time or quasi real time. P5,l6: two commas (,,) probably a lacking sentence. P5,l11-12: too generic: unrealistic? Manual check? What is the algorithm (objective method) behind it? P5,l20: "IDP are statistically too high or too low", this sentence is generic. P5,l21: "too high" - "slightly overestimate intensity" the two sentences appears in contradiction. P5,l29: Mean is not very appropriate for int. estimation, if you follows intensity degrees definition you will find, for example, an evaluation of percentage of people observing such effect which it is associated an int. Value. P5,l30: statistical errors: unhappy terms in this contest, what does it mean? Probably an error component too high. P6,l19-20: check language. P6,l22: which is the time length of the catalogue? Otherwise the sentence has a poor meaning. P7,l21: Agencies are national but some collect also international data: whole set of data is international. P7,l22-28: was a statistical test conducted to assess spatial variability and localization precision of data? I think it should be worth to quantify it instead to give a qualitative evaluation based on personal opinion. P7, l35-36: "The impact of differences between the institutional questions on the intensity scale remains very low " How can you state it? Statistical

analysis? Referenced results? Please specify. P7, l37: Same as above: what do you mean with "spatial variability" ? Do you have a quantification of it to assert the differences among different data sources? P8,l1-2: I do not understand why merging data removes spatial variability. Merging different data sources increases variability. P10,l16-20: Here the hypothesis of normal distribution is not correct because intensity data are strongly conditioned by radial geometry and log distance attenuation laws. Explanation does not seem reasonable and well argumented. Intensity data for a whole macroseismic field are not supposed to be normally distributed being influenced by the aforementioned factors plus undersampling and dependence of estimation error to intensity. P10,l21-22: This sentence is obvious: it is as to say that the whole is more than a part. P10,l31-34: Why not trying to statistically correct this discrepancy? For example making a correlation among different data set, looking for corrective coefficients. P10,l38: It is known that data deriving from non permanent effects are strongly based on compilers' memory: it could be useful to search for a dependence of answers errors with compilation times. P11,l1-2: This sentence is not clear, entries are generally not random in time. In fact they follow a time decay law (resembling a sort of Omori law) modulated by day-night cycle. In space the distribution could be compared with spatial citizen density distribution. P11,l4-10: Intensity spatial distribution is based on qualitative evaluation and results are expressed with vague, colloquial terms, as example "far from circular radiation".A quantitative approach could be based on analytic comparison with experimental data and isotropic fitting. P11,l13-14: But it is biased on radial areal increment due to polar distribution. Moreover IAR derived from an equal area is not sufficient to assess unbiased results. There is a need of further analysis, for example comparing attenuation relations from each agencyseparately. P11,l36: I find a contradiction: on one side the authors find that their data are at the epicentral zone characterized by lower intensity comparing with suitable attenuation law (Atkinson and Wald), on the other side they state that first 50 km attenuation is due to fast energy decay of seismic energy from the source. It could be explained why fast decay did not affected attenuation laws. P13,l29-30: this part is an example of uncertainties

stemming from not considering of not felt individual reports, in fact the authors decided to reduce intensity from III to II basede on reasonable consideration. Not considering not felt percentage is a weak point of the investigation. P13,l36: An other example of qualitative analysis: the Ramsgate intensity distribution shows a WNW-ESE orientation (Fig. 10), can you quantify/justify this sentence? P16,l20-30: the comparison between intensity and depth of geological structures could be done in more qualitative way, for example performing a correlation between intensity residuals and structures. P17,l1-3: depth differences of the two earthquakes is small taking into account depth estimation uncertainty. P18,l3: "why" repeated: eliminate it. P21,l46 correct family name is De Rubeis

---

## Referee Comment (RC3) · Anonymous Referee #3 · 13 Jan 2017

I won't repeat the points that Remy Bossu and Valerio de Rubeis stressed very nicely in their reviews, but will stress out just a few things that I consider crucially important:

DYFI is the USGS online questionnaire; it is very famous and very popular. But, it's not the basis of the majority of European national online questionnaires. The tradition of collecting macroseismic data in organised way is old and rich in European countries. Almost each of them has developed its own national questionnaire, based on the scale that was used locally, as well as including details that were of importance.

It is definitely not written in EMS guidebook that one should decrease the intensities from the observers in third and fourth floors by one intensity value.

[Figure]

To exclude EMSC questionnaires because the intensities were not in accordance to the average values of other institutes is definitely not scientifically correct.

But my main problem is the following: evaluating the intensity for some locality means to collect all the data about earthquake effects in that town or village and evaluate them together in order to obtain the intensity value for the said locality. It is not correct to assign individual intensity values to each observation and then recalculate the intensity following some rule. This is the only way to be sure that the intensity value is correct, and to obtain reliable seismic history of the settlement. Here, however, no one seems to care much about the earthquake effects described in questionnaires; lot of effort is put into fiddling with the already calculated intensities instead.

What is the use of this? I can give it a benefit of being handy for showing the rough outline of the intensity field soon after the earthquake. But this can not be a tool to really study a transfrontier earthquake. There is much more behind each coloured circle on the map than just playing with the grid size.

I feel bad for writing all this, as it is obvious that there is a great effort behind this work; I would sincerely recommend the authors to reconsider what they did and to adjust the already prepared framework to deal with the intensity data again, this time using the proper methodology.

---

## Author Comment (AC1) · 11 Feb 2017

**Response to review of R. Bossu**

- *The authors have been using "Did you Feel it ?" not only to present the tool developed and operated by the USGS but also more generally for "Internet macroseismic data". (One should note that the title do use the latter). I find this confusing. I believe this distinction should clearly appear in the text. DYFI was the very first online macroseismic tool, several institutes have implemented the same questionnaires, but others have developed their own approach.*

We agree with this comment. To clarify the text we now only use 'DYFI?' if we specifically refer to the USGS macroseismic inquiry. Reference to any other questionnaire is made by using "internet macroseismic data", such as indicated in the title.

- *My second issue is about the description of the data used. I believe that a description of the methodology for each data provider (perhaps in appendix) would be useful.*

We agree that this comparison was overdue. We added a table (see below, will be added as a Supplement) with comparison of the different questionnaires used in this study to the supplementary data. In this table, we checked which question (40!) in the different questionnaires of the seven institutes (BGS, ROB-BNS, NRW-GD, BCSF, EMSC & USGS) do (not) overlap. The table is interesting: most questions are rather similar (i.e. person's situation, perception and experience of the earthquake) but each questionnaire does have its specifics and no two questionnaires are completely alike. This table also revealed why the NRW-GD has no intensity I values in their database: they don't have a Q13: "have you felt the earthquake".

- *Is it a questionnaire (or thumbnails),*

Only the EMSC and BSCF currently provide thumbnails.

- *How are the locations determined (zip code, geocoded full address, nearest city which was an option at EMSC when eyewitnesses declined to provide their full address)*

At the end of the comparison table we added the locations procedure (if known).

- *and how the intensity is assigned.*

We cited to the proper references in case the intensity algorithm is known and mention them in section 4 in the data description.

- *The paper makes a very strong assumption (last sentence of page 2) that intensities may slightly differ from one country to the other (due to differences in questionnaire and/or intensity assignment procedures). Some of the data presented in this paper contradict this statement: the EMSC macroseismic data derived from questionnaires had to be excluded because they differ too much from the other datasets. (For information, these excluded intensities had been assigned by an algorithm developed by one of the father of the EMS98 scale). What I want to stress here is that there is no reference to such a statement and my own experience, or recent by Hough, Martin et al comparing macroseismic datasets for Ghorka earthquake do not support it. This is probably too much work to fully address this issue, but the assumption that differences in intensity from one country to another are slight should be made clear and explicit. A consequence of the previous point is that the methodology is about the spatial grouping of different Internet macroseismic data only.*

We agree with this comment and reformulated the text in the introduction and in sections 2 and 4. The comparison table that is added to the Supplement shows substantial differences between the questionnaires. This comment **is one of the key points of the paper**: namely that a profound review of all questionnaires and their impact on the intensity scale is needed in Europe. Currently, the intensity determination procedures are not always transparent, leading to different results for the same area. This is the painful truth in Europe.

To explore the influence of the questionnaire, we generated institutional IARs through the datapoints for the Goch and Ramsgate earthquakes (see review De Rubeis). Some institutional IARs (ROB-BNS, BSCF, USGS, EMSC questionnaire) are rather similar despite the different questions. Other (NRW-GD, EMSC thumbnail) differ strongly from the main IAR. We added this observation to the discussion and to the perspectives at the end of the paper, rather than including it as an observation. Currently, the intensity procedures are not always known, leading to different results in the same area.

We changed section 4.2 as follows:
*"The tradition of collecting macroseismic data in an organised way is old and rich in European countries. Table S1 (Supplement) compares all questions in the different institutional questionnaires. The questions concern typical effects on the person's situation when the earthquake occurred, the perception and experience of the earthquake and earthquake effects on furniture, buildings and the environment. Each questionnaire originates from an historical form developed for a local intensity scale or is modified after the pioneering online questionnaire of Wald et al. (1999). Notwithstanding much overlap between the questions, no two questionnaires are alike. The impact of these differences*

*on the intensity scale is unknown and might be present, such as recently shown by Hough et al. (2016) on macroseismic datasets for the Nepal Ghorka earthquake."*

- *In the second paragraph of the introduction, the EMSC is presented alongside the national institute while it works similarly to the USGS "Did you feel it". This seems to indicate there is no transfrontier and international internet macroseismic data collection in Europe or that EMSC works at national level, which is not the case.*

This was a mistake. EMSC has now been added to the international institutes

- *EMSC does not request not felt response from volunteers. The LastQuake app send notification after felt earthquake to people in the area and some of them may react to this notification by sharing their testimony*

Ok, this sentence has been modified adding the information above.

- *First sentence of the conclusion is inaccurate. Transborder macroseismic maps exist in Europe at EMSC. The challenge is to create a denser and possibly more accurate one by merging national datasets.*

Ok, this sentence has been modified

- *Third paragraph of the conclusion: the paper does not demonstrate "strongly improves the quality of real time intensity evaluation of individual agencies". Neither intensity assignment nor real time processing is covered in this article*

Although we do not cover any real-time processing, the work in this paper aimes to set an example how data **could** be shared in Europe in quasi real-time. Hence, any statements on "real-time" processing have been deleted from the paper but we want to stress in the conclusions that generating dense transfrontier maps using national macroseismic data in Europe stays problematic. Only after a careful analysis of the different available questionnaires and their impact on the intensity scale, we can exchange and process intensity data in real-time.

**COMPARISON OF MACROSEISMIC QUESTIONNAIRES USED IN VAN NOTEN ET AL. (2017), Solid Earth**

| Nr. | Questions | Possible answers | ROB-BNS | BCSF | KNMI | NRW-GD | BGS | EMSC | USGS |
|---|---|---|---|---|---|---|---|---|---|
| 1 | Date and Time | | yes | yes | yes | yes | yes | yes | yes |
| 2 | Street, Address | | yes | yes | yes | yes | yes | yes | yes |
| 3 | Zip code, City, Country | | yes | yes | yes | yes | yes | yes | yes |
| **PERSON'S SITUATION WHEN THE EARTHQUAKE OCCURRED** | | | | | | | | | |
| 4 | How many times have you felt an earthquake ? | 1st, few times, often | no | no | no | no | no | yes | no |
| 5 | What was your situation during the earthquake? | No answer / Inside / Outside / In a stopped vehicle / In a moving vehicule / other | yes | yes (less precise) | yes | yes | yes | yes | yes |
| 6 | What was your situation during the earthquake? | Other: church tower / electricity mast / scaffolding | no | no | no | Yes | no | no | no |
| 7 | If you were inside, please select the type of building or structure | No building, Family home, Apartment building, Office building/school, Mobile Home with permanent foundation / Trailer without fundation | yes | yes | yes | Partly | yes | no | no |
| 8 | At what floor where you? | Floor number | yes | <2, 2/3, 3/4, >=5 | yes | No | 1,2,3,4-8,>8 | no | no |
| 9 | Type (wood, brick, etc.) of the building | wood / brick / concrete / loam / … | yes | yes | yes | yes | Yes | no | no |
| 10 | Height (in floors) of the building | specify | yes | yes | yes | no | yes | no | no |
| 11 | Type of activity during event | Standing, sitting, lying, walking, kneeling, sleeping | no | yes | no | yes | yes | yes | no |
| 12 | Swinging effect of the respondent | Standing up, swaying, fell | no | yes | no | yes | no | yes | no |
| **PERCEPTION OF THE EARTHQUAKE** | | | | | | | | | |
| **13** | **Did you feel the earthquake?** | no / yes | yes | yes | yes | no | yes | yes | yes |
| 14 | Were you asleep during the earthquake? | no / yes, didn't get up / yes, did get up | yes | yes | yes | yes | yes | yes | yes |
| 15 | What best describes any sound you heard? | no sound / rumblinb / roaring / explosion | no | yes | no | yes | yes | yes | no |
| 16 | Did you hear a noise? How loud ? | no / yes, slight, loud noise | yes | yes | no | no | yes | yes | yes |
| 17 | Did you hear church bells ? | no / yes | no | no | no | yes | no | yes | no |
| 18 | Did other persons nearby feel the earthquake? | I don't know, nobody nearby/some felt it, others not/most felt it, others not/(almost) everone felt it | yes | no | yes | yes | yes | yes | yes |
| 19 | Have you felt shocks before or after, if so how long/many | Specify | no | YES (last case observations) | no | yes | no | no | no |
| **YOUR EXPERIENCE OF THE EARTHQUAKE** | | | | | | | | | |
| 20 | How would you best describe the ground shaking? | weak / mild / moderate / strong / violent | yes | yes | yes | no | yes | yes | yes |
| 21 | How would you describe the earthquake shaking | vibrating / trembling / swaying / impact / rolling | no | yes | no | yes | yes | yes | no |
| 22 | About how many seconds did the shaking last? | Specify | yes | no | yes | yes | no | no | no |
| 23 | How would you best describe your reaction? | no reaction / very little reaction / excitement / (somewhat, very, extreme) frightened | yes | yes | yes | partly | yes | yes | yes |
| 24 | How did you respond? | No action / moved / cover / ran outside | yes | yes | yes | | yes | no | yes |
| 25 | Was it difficult to stand or walk? | no / yes (difficult, fallen, forcibly thrown) | yes | yes | yes | no | no | no | yes |
| 26 | Did you notice the swaying or swaying of doors, windows or free-hanging objects? | No answer, did not look / yes (slight/violent swinging) | yes | yes (only objects) | Yes | Yes, only checkbox | yes | yes | yes |
| 27 | Did you notice creaking or other noises? | No answer, paid no attention / yes (slight/loud noise) | yes | yes | yes | Yes, only checkbox | no | no | yes |
| 28 | Did objects rattle, topple over, or fall off shelves? | No answer, no shelves / Yes: slight/loud rattle - few toppled - few/many/everything fell off | yes | yes | yes | Yes, only checkbox | yes | yes | yes |
| 29 | Did pictures on walls move or get knocked askew? | No answer, no furniture / no / yes | yes | yes | yes | Yes | yes | yes | yes |
| 30 | Did any furniture or appliances slide, tip over, or become displaced? | No answer, no heavy appliance / no / yes, some contents fell, shifted few cm, shifted a foot (30 cm), overturned | yes | yes | yes | Yes, only checkbox | yes | yes | yes |
| 31 | Was a heavy appliance (refrigerator or range) affected? | No answer, no furniture / no / yes | yes | yes | yes | no | yes | no | yes |
| 32 | Moving liquids, oscillation / overflow of liquids in bowls? | no / yes / don't know | no | yes | no | yes | yes | yes | no |
| 33 | Did trees / bushes swing ? | no / yes / don't know | no | no | no | yes | yes | yes | no |
| 34 | Were free-standing walls or fences damaged? | No answer, no walls / no / yes, some were cracked/partially fell/completely fell | yes | no | yes | no | no | no | yes |
| 35 | If you were inside, was there any damage to the building? Check all that apply: | | | | | | | yes | |
| | *No damage* | | yes | no | yes | no | yes | yes | yes |
| | *Hairline cracks in walls* | | yes | yes (+quantity) | yes | Yes, only checkbox | yes | | yes |
| | *A few large cracks in walls* | | yes | yes | yes | | yes | | yes |
| | *Many large cracks in walls* | | yes | yes | yes | | yes | | yes |
| | *Ceiling tiles or lighting fixtures fell* | | yes | yes | yes | no | no | | yes |
| | *Cracks in chimney* | | yes | yes | yes | yes | yes | | yes |
| | *One or several cracked windows* | | yes | no | yes | yes | yes | not specified | yes |
| | *Many windows cracked or some broken out* | | yes | yes | yes | yes | yes | | yes |
| | *Masonry fell from block or brick wall(s)* | | yes | yes (+quantity) | yes | no | yes | | yes |
| | *Old chimney, major damage or fell down* | | yes | yes, but without age distinction | yes | Yes, no age distinction | Yes, no age distinction | | yes |
| | *Modern chimney, major damage or fell down* | | yes | yes (+quantity) | yes | | | | yes |
| | *Outside wall(s) tilted over or collapsed completely* | | yes | no | yes | no | no | | yes |
| | *Separation of porch, balcony, or other addition from building* | | yes | no | yes | no | no | | yes |
| | *Building shifted over foundation* | | yes | no | yes | no | | | Yes |
| 36 | Did the roof collaps? | Total / part (quantity) | no | yes | no | yes | House partly or completely collapsed ? | no | no |
| | Did any poles or storeys collapse? | yes / no | no | yes | no | no | | no | no |
| | Cracks at joints, poles, wall corners? | specify | no | yes | no | no | | no | no |
| | Did parts of walls or the facade collapse? | yes / no | no | yes | no | no | | no | no |
| 37 | Environmental effects | Ground cracking / landslided / waving ground movement | no | no | no | yes | yes | yes | no |
| 38 | Unusual animal behaviour | No / Yes, pets, farms animals, no animals nearby | no | no | no | yes | yes | yes | no |
| **Are THUMBNAILS provided?** | | | no | **yes** | no | no | no | **yes** | no |

| Type of intensity maps? | | ROB-BNS | BCSF | KNMI | NRW-GD | BGS | EMSC | USGS |
|---|---|---|---|---|---|---|---|---|
| | Zip code map | yes | yes | yes | yes | yes | yes | yes |
| | Geocoded "boxes" maps | yes | no | no | no | no | no | yes |
| | Nearest city maps | no | no | no | no | no | yes | yes |

---

## Author Comment (AC2) · 11 Feb 2017

**Response to the review of V. De Rubeis**

**General questions:**

- *Did you take into account the percentage of not felt to asses an intensity degree, as macroseismic scale recommends?*

We do take the not-felt reports into account in our analysis, which avoids the intensity in the cells being too high. To clarify the datasets we used, we added the amount and percentage of not-felt reports in tables 1 and 2. The total percentages of not-felt reports in the geocoded dataset are very low (663/8491, 8% for Goch and 260/3953, 7% for Ramsgate). We do agree that working with percentages of not felt responses would lead to a more robust intensity analysis. However, the percentages of not-felt reports within the grid cells are very low (mean~0.03%). As we have no voluntary base such as INGV and don't have an observers network (we do need this in the future) we cannot assess the not-felt percentages reliably and cannot make a more robust interpretation. We added this remark to section 4.3.6

It is clear from the intensity maps that working with cells has benefits above working with ZIP codes as more not-felt grids are shown.

- *Can you try to compare attenuation laws for each data sources?*

This is a very good idea. Yes we can. We ran our attenuation algorithm again through all individual data sources. Figure 7A & 11A show the results of this analysis. Combined with the questionnaire comparison table (see review Bossu) this plot shows a strong difference for the NRW-DG data (overall higher intensity) and the EMSC_thumbnail data (higher epicentral intensity) for Goch. The shape of the IAR's of all other data sources have a rather similar shape in the first 100 km's, showing that although the individual questionnaires might differ, the intensity results are similar!

For the NRW-GD this analysis confirms that 1) absence of not-felt reports drives their IAR up and 2) their intensity results cannot be used in the overall analysis (see further). For the Ramsgate earthquake we limited the IAR's to those areas in which the data source gathered enough reports. BGS individual data is not available so we plotted the BGD grid data which show higher, but more accurate intensities (see comment review #3).

[Figure]

Figs. 11A & 7A

[Figure]

Figs. 7B

**Specific questions of V. De Rubeis**

- *P2, l3: change real time to quasi real time.*

ok

- *P2,l11: not complete. Pattern depends on source also, change the sentence like: Concentric pattern of intensity decay is only a theoretical very first approximation, which may serve only to indicate seismic epicentral best location.*

Thank you for this suggestion. Sentence changed.

- *P2,l25 and l34: explain the meaning of real time or quasi real time.*

As there might be a confusion with these sentences, the paragraph has changes as follows:
*"Despite this high number of inquiries only a minority of these national institutes, i.e. BGS, INGV, BCSF, SED, IGN, ROB-BNS (see appendix A), calculate and map intensities from the people's submissions online and update a macroseismic map in quasi real-time (a small amount of time is still needed to process the data)."*

- *P5,l11-12: too generic: unrealistic? Manual check? What is the algorithm (objective method) behind it?*

We did not explain this in the paper. First we analysed all data and constructed an IAR through the data. Afterwards if individual intensities are higher than +III intensity units above the IAR at larger distances (e.g. for Goch > 150 km), we delete these high intensities as they are not very reliable. This procedure should be similar to the procedure used at the INGV. Afterwards we re-computed the IAR, now without these high intensities.

- *P5,l20: "IDP are statistically too high or too low", this sentence is generic.*

We deleted this sentence.

- *P5,l21: "too high" - "slightly overestimate intensity" the two sentences appears in contradiction.*

We changed this sentence to avoid this contradiction.

- *P5,l29: Mean is not very appropriate for int. estimation, if you follows intensity degrees definition you will find, for example, an evaluation of percentage of people observing such effect which it is associated an int. Value.*

We agree with this comment, but we cannot do better than taking the mean. Apart from the USGS, the data provided by the institutes were only individual intensity data. We do not have the detailed answers to each individual question, which complicates to assess a proper EMS98 intensity value. See our reply to R#3 for a detailed answer.

- *P5,l30: statistical errors: unhappy terms in this contest, what does it mean? Probably an error component too high.*

Indeed, we rewrote this sentence.

- *P6,l19-20: check language.*

Ok, this sentence has been rewritten: The Goch epicentre is located 6 km ± 2.7 km NE of the surface trajectory of the SW-dipping Viersen fault. Given the 10 km source depth, it is thus not possible to attribute the Goch earthquake to the Viersen fault.

- *P6,l22: which is the time length of the catalogue? Otherwise the sentence has a poor meaning.*

The ROB and BNS instrumental catalogues start from 1985 and 1975, respectively.

- *P7,l21: Agencies are national but some collect also international data: whole set of data is international.*

OK, we rewrote this sentence following your suggestion.

- *P7,l22-28: was a statistical test conducted to assess spatial variability and localization precision of data? I think it should be worth to quantify it instead to give a qualitative evaluation based on personal opinion.*

No statistical test was performed. The high precision of the geocoded data (ROB-BNS, KNMI) is provided by the quality factor after the Google geocoding. Given the small error on the location precision (1.11 km for USGS; 0.11 km for NRW) we don't see the need of performing a statistical test. In 100 km$^2$ cells the worst case scenario that can happen is that a few points might fall in the next cell, shifting the intensity.

- *P7, l35-36: "The impact of differences between the institutional questions on the intensity scale remains very low " How can you state it? Statistical analysis? Referenced results? Please specify.*

Please see our detailed reply to Bossu who asked a similar question.

- *P7, l37: Same as above: what do you mean with "spatial variability" ? Do you have a quantification of it to assert the differences among different data sources? P8,l1-2: I do not understand why merging data removes spatial variability. Merging different data sources increases variability.*

This is true, due to the different datasets, different precisions of location and the different intensity procedures, merging data will increase the variability of the total dataset. We changed this as follows:

"Merging different macroseismic data sources allows having more data and provides a denser spatial coverage of the area over which the earthquake was felt. It however also increases the variability of the dataset due to increased uncertainty in the respondents' locations as different location procedures are used by the agencies. "

- *P10,l16-20: Here the hypothesis of normal distribution is not correct because intensity data are strongly conditioned by radial geometry and log distance attenuation laws. Explanation does not seem reasonable and well argumented. Intensity data for a whole macroseismic field are not supposed to be normally distributed being influenced by the aforementioned factors plus undersampling and dependence of estimation error to intensity.*

This is a true comment. We deleted this argument.

- *P10,l21-22: This sentence is obvious: it is as to say that the whole is more than a part.*

The original sentence was badly phrased, sorry. What we want to state is that for the Goch earthquake, the sum of responses submitted to the national institutes is far larger than the sum of those which arrived at international agencies. This shows the importance of the national institutes in collecting macroseismic data, particularly for moderate earthquakes!

- *P10,l31-34: Why not trying to statistically correct this discrepancy? For example making a correlation among different data set, looking for corrective coefficients.*

We actually tried this by comparing the intensity values for those cells in which NRW-GD and ROB-BNS have common data. It is clear that the NRW-GD overestimates intensities. There are several ways to correct for this error but they all would depend on the chosen fit through the correlated data. As we have no idea in which way the error is consistent (linear, powerlaw, exponential,..) we decided to not correct the data and only use the NRW-GD data as "felt". Correcting the data would increase the uncertainty. In almost the entire region where NRW has data, we have overlapping responses from other institutes.

[Figure]

Fig: black points = NRW-ROB correlation for common cells. Blue cells = "correcting" the data to the 1:1 line by inverting the linear interpolation between the black points.

- *P10,l38: It is known that data deriving from non-permanent effects are strongly based on compilers' memory: it could be useful to search for a dependence of answers errors with compilation times.*

See response to next question.

- *P11,l1-2: This sentence is not clear, entries are generally not random in time. In fact they follow a time decay law (resembling a sort of Omori law) modulated by day-night cycle. In space the distribution could be compared with spatial citizen density distribution.*

This was indeed wrongly formulated in the text. We added a figure in the Supplement explaining the response submission rate of the Goch and Ramsgate earthquakes to the seismology.be website (ROB-BNS). Both curves show that submissions were indeed not random in time. See explanation in the Supplement.

Also in space this is not random but both controlled by the population density. As we discuss the distribution and population density in the discussion, we removed this sentence here.

- *P11,l4-10: Intensity spatial distribution is based on qualitative evaluation and results are expressed with vague, colloquial terms, as example "far from circular radiation". A quantitative approach could be based on analytic comparison with experimental data and isotropic fitting.*

The expected circular behaviour of the Goch intensity pattern is altered by the local geology. We highlight this in the discussion. In section 4.4.6 we only want to describe the shape of the intensity distribution. We understand your concern so reformulated this sentence to have a more precise description: "The Goch grid cell intensity pattern distribution deviates from an expected circular pattern as intensity cells are absent in the SW quadrant (Fig. 6)."

- *P11,l13-14: But it is biased on radial areal increment due to polar distribution. Moreover IAR derived from an equal area is not sufficient to assess unbiased results. There is a need of further analysis, for example comparing attenuation relations from each agency separately.*

The areal increment is best visible in the datasets of KNMI and NRW. We calculated the institutional IARs (Fig. 7A) of the institutes from both the individual and grid cell data. We added these IARs to Fig. 7A but left them out of Fig. 7B to not complicate this figure. But the trends are the same. The IARs show that the KNMI, ROB-BNS, USGS and EMSC (qu) all result in a similar distance decay. This confirms that although the questionnaires might differ, the gathered intensity data are rather similar and the radial areal increment due to polar distribution does not affect the IAR. We added this aspect to the paper.

- *P11,l36: I find a contradiction: on one side the authors find that their data are at the epicentral zone characterized by lower intensity comparing with suitable attenuation law (Atkinson and Wald), on the other side they state that first 50 km attenuation is due to fast energy decay of seismic energy from the source. It could be explained why fast decay did not affected attenuation laws.*

In fact this is a local versus regional effect. In the epicentral area, the local site effect (deep bedrock) attenuates the intensity and lowers the IAR with respect the A&W (2007) attenuation law. With increasing distance, a larger area is considered and intensity data above different geological units is gathered. This smooths the IAR into a 'normal' situation where it approaches the A&W07 CEUS law. We deleted the suggestion that this is related to fast energy decay of seismic energy from the source and added the explanation above to the discussion.

- *P13,l29-30: this part is an example of uncertainties stemming from not considering of not felt individual reports, in fact the authors decided to reduce intensity from III to II based on reasonable consideration. Not considering not felt percentage is a weak point of the investigation.*

See main answer above.

- *P13,l36: Another example of qualitative analysis: the Ramsgate intensity distribution shows a WNW-ESE orientation (Fig. 10), can you quantify/justify this sentence?*

Yes we can. We performed azimuthal analysis through all cells for which we have intensity data. Figure S1 in below and provided in the Supplement shows the results of this analysis.

The polar plot shows the mean azimuth calculated through all response cells on the continent and in the United Kingdom. **On the continent**, the mean orientation (red line, brown area = ± 1σ) of the felt distribution is 111.5°, corresponding to an ESE orientation with respect to the epicentre. The ESE-oriented felt distribution clearly deviates from the mean azimuth (132.7°, grey area = ± 1σ) derived through all available cells, i.e. the response potential on the continent. This orientation shows that the distribution is not a population, nor an "emerged land vs sea (blue dots)" effect.

**In the UK**, the response distribution is more widespread (1σ between 260° to 350°) but the azimuthal mean is 305.6° ~WNW. This larger spread is likely related to the dispersing occurrence of the Anglo-Brabant Massif in the subsurface (see Fig. 12 in the paper).

This polar plot proves a WNW-ESE orientation of the felt distribution of the 2015 Ramsgate earthquake, an orientation clearly following the tectonic structure of the Anglo-Brabant Massif.

We added to the text *"Numerical azimuthal analysis of the grid cell centroids relative to the epicentre (see Fig. S1 in the Supplement) quantitatively proves that the Ramsgate intensity distribution is clearly WNW-ESE oriented with a mean circular azimuth of 112° on the continent and 306° in the UK, respectively."*

[Figure]

**Figure S1** added to the Supplement.:

Blue plots = cells within the North Sea and English Channel

Coloured cells = intensity data.

Shaded areas = +- 1sigma around the circular mean.

*P16,l20-30: the comparison between intensity and depth of geological structures could be done in more qualitative way, for example performing a correlation between intensity residuals and structures.*

In order to apply intensity residuals, we need to have an established attenuation model; however, such an IAR constructed from internet data is currently lacking in our regions. We are not convinced that constructing intensity residuals from the Goch IAR would confirm the geological structures. Whatever existing IAR we would take (either from historical analysis or internet data) it would always be a rough approximation, especially because of the different geology over the whole area. You are right that this needs be done in the near future to perform a more qualitative analysis, but only after evaluating all felt earthquakes in our database and not only with the IAR of the Goch or the Ramsgate earthquakes.

*P17,l1-3: depth differences of the two earthquakes is small taking into account depth estimation uncertainty.*

After revision of the seismic data we could reduce the depth uncertainty to 1.2 km for Goch and 3 km for Ramsgate. Goch was thus still shallower than Ramsgate indicating that a depth effect is still present in the macroseismic data. This depth difference is clearly reflected in the difference in epicentral distances over which the Goch and Ramsgate events were felt. Ramsgate (max ~ 360 km) > Goch (max ~ 250 km). This is clear in the discussion and we did not change anything.

---

## Author Comment (AC3) · 11 Feb 2017

**Response to Anonymous Referee #3**

- *I won't repeat the points that Remy Bossu and Valerio de Rubeis stressed very nicely in their reviews, but will stress out just a few things that I consider crucially important: DYFI is the USGS online questionnaire; it is very famous and very popular. But, it's not the basis of the majority of European national online questionnaires. The tradition of collecting macroseismic data in organised way is old and rich in European countries. Almost each of them has developed its own national questionnaire, based on the scale that was used locally, as well as including details that were of importance.*

Thanks to indicate this error. We rewrote the introduction and section 4.2 mentioning the information above. More information can be found in our reply to R. Bossu.

- *It is definitely not written in EMS guidebook that one should decrease the intensities from the observers in third and fourth floors by one intensity value.*

This is a correct comment. –I intensity decrease for 3&4 floors is not 'literally' advised in the EMS-98 guideline. But allows us to explain why we do this rescaling (this has also been rephrased as such in the paper):

EMS98 says : *"One special case is the situation where the only reports are from tall buildings, because the shaking was so weak that it was only perceptible on the upper floors of such structures. This sort of datum is typical of intensity 2. »*
This is insufficient information given by the EMS98 scale. If you are near an epicentre with intensity e.g. III, IV or V and you only receive a few online answers from people living in a high-rise building, then this value II is obviously false. Only a sufficient amount of data (see also further our reply to your main concern) can bring a reliable intensity assessment, which is obviously not the case with internet data.

EMS98 also advises (P.29) :*"It is well-known that people in upper storeys are likely to observe stronger earthquake vibration than those in lower storeys…. Various practices, such as reducing the assigned intensity by one degree for every so many floors, have been suggested, but never found general favour."*
Here again, the EMS provides insufficient guidelines. We know that changing the intensity is not a perfect approach because each building has its own frequency, different floor height, etc… but we do think that adjusting intensities for higher floor responses is better than doing nothing. The experience of the BCSF with internet data proves that this adjustment is important (in particular for big cities) and that this needs to be integrated in order not overestimate the intensity value in comparison with classic method of estimation.
The recommended practice by EMS98 is to discount all reports from observers higher than the fifth floor when assigning intensity. So reducing the assigned intensity by –I intensity only for floors 3 and 4 is not strictly prohibited by EMS98 scale.
This is another point that the EMS98 needs to take into account in the future and which was also addressed by Sbarra et al. 2012.

- *To exclude EMSC questionnaires because the intensities were not in accordance to the average values of other institutes is definitely not scientifically correct.*

OK, we added the EMSC questionnaires to our analysis which slightly increased the number of responses used in this paper (see tables 1 and 2 and Fig. 7A, see comment De Rubeis). The mean IAR of the Goch earthquake (fig. 7A) shows that the intensities derived from thumbnails and questionnaires are slightly different. Intensity analysis by using thumbnails tends to give higher intensities in the epicentral area than the questionnaires. We don't know why. Perhaps people tend to choose a higher intensity if they know they are close to the epicentre… Finding an appropriate answer to this question is unfortunately beyond the scope of the paper. Apart from the NRW-GD, all institutes show rather similar IAR's. See comments in the reply to V. De Rubeis.

- *But my main problem is the following: evaluating the intensity for some locality means to collect all the data about earthquake effects in that town or village and evaluate them together in order to obtain the intensity value for the said locality. It is not correct to assign individual intensity values to each observation and then recalculate the intensity following some rule. This is the only way to be sure that the intensity value is correct, and to obtain reliable seismic history of the settlement. Here, however, no one seems to care much about the earthquake effects described in questionnaires; lot of effort is put into fiddling with the already calculated intensities instead. What is the use of this? I can give it a benefit of being handy for showing the rough outline of the intensity field soon after the earthquake. But this cannot be a tool to really study a transfrontier earthquake. There is much more behind each coloured circle on the map than just playing with the grid size.*

We understand your concern and agree that evaluating earthquake effect percentages is the recommended method to assign intensities. However, let us explain why we choose a more simple and hence more rough methodology. The merged institutional intensity data of the Goch and Ramsgate that we have in our possession did not allow applying the recommended procedure: EMSC, NRW-GD and KNMI only gave us a list with individual intensities and coordinates, without the answers to the specific questions. The grid cell intensity from the BGS was taken from their website. EMSC also provided clustered data. From the USGS, BSCF and ROB-BNS detailed answers are available. This means that we have a part of the answers and hence simply cannot do better with the provided data than merging individual intensities. We prefer to perform a rough intensity analysis on the whole macroseismic field rather than applying a statistically-correct analysis on a subset of the data.

The current situation with individual testimonies collected online is also different than that with communal testimonies gathered from classical inquiries. Online macroseismology provides an alternative way to obtain a preliminary intensity value with only few testimonies. Certainly, online data does not replace, nor is invented to replace/improve the classical method but we must continue working in this direction to obtain new methodologies that can be applied in quasi real time based on few data. If we would strictly respect the EMS98 guideline, then we can stop with online macroseismology by individual internet enquiries because the data collected in cities or in squares are often not sufficient and representative due to their low content, not in the UK, not on the continent. Boatwright and Phillips (SRL, 2017) recently provided a new methodology to recalibrate intensities taking into account population density and response reaction. This might resolve the population issue in the future but it is not the same as carefully evaluated communal reports. Even if the recommended procedure would be followed, in numerous cases, the number of individual data collected in cells is often insufficient to realize a representative statistical processing of the intensity. To make a statistical analysis of earthquake effects with only few testimonies in a square is perhaps not better than the thumbnail averaging.

Our grid cell analysis is indeed a preliminary result but it proposes a new solution to obtain a rapid severity of shaking and which can potentially be used in quasi real time. That's it, not more. We performed this study to evaluate a methodology, it's pros and cons to create dense macroseismic maps by merging data. And in fact **the results are not so bad**: e.g. for Goch, the same site effects are confirmed as with other earthquakes (e.g. Roermond 1992) that were analysed in the same region with the classic communal methodology. After this paper, we agree it will be necessary to make a survey to obtain a more consolidate result conform to EMS98 procedure.

To conclude, you are right that we cannot assign EMS98 values by grouping intensities in cells. We changed this statement in the paper as it is not certain that a strong and certified EMS98 intensity value can be determined. The used intensities correspond to "individual severity of shaking" which is determined by following as much as possible the EMS98 guideline. Communal and field inquiries using the classical procedure are still needed complementary to individual internet data.

We added the following text to the paper to clarify our statement and methodology:
*"Intensity degree definitions recommend to evaluate the percentage of people observing an earthquake effect, of which each effect is associated with an intensity value. Grouping the intensity values of a large number of responses in ZIP code areas leads to a robust intensity assessment. Unfortunately, the European macroseismic fragmentation complicates applying this procedure. The transfrontier Goch and Ramsgate macroseismic data that were provided to us by the (inter)national agencies only contained 'individual' intensities and not the detailed answers to the questions. Hence, the recommended procedure could not be followed. Moreover, in numerous cases, the number of individual data collected in cells is often insufficient to realize a representative statistical processing of the intensity. For all these reasons we decided to calculate the mean intensity of all (geocoded if possible) individual intensities within a cell. This grid cell intensity is an approximation of the intensity field and provides no certified EMS-98 value but it provides a solution to obtain a rapid severity of shaking after merging data."*